



# Multivariate state and parameter estimation with data assimilation on sea-ice models using a Maxwell-Elasto-Brittle rheology

Yumeng Chen[1], Polly Smith[1,†], Alberto Carrassi[1,2], Ivo Pasmans[1], Laurent Bertino[3], Marc Bocquet[4], Tobias Sebastian Finn[4], Pierre Rampal[5], and Véronique Dansereau[6]

[1]Department of Meteorology and National Centre for Earth Observation, University of Reading, Reading RG6 6ET, UK
[2]Department of Physics and Astronomy "Augusto Righi", University of Bologna. Bologna, Italy
[3]Nansen Environmental and Remote Sensing Center, 5007 Bergen, Norway
[4]CEREA, École des Ponts and EDF R&D, Île-de-France, France
[5]Institut de Géophysique de l'Environnement, Université Grenoble Alpes/CNRS/IRD/G-INP, CS 40700, 38 058 Grenoble CEDEX 9, France
[6]Université Grenoble Alpes, CNRS, Grenoble INP, Laboratoire 3SR, Grenoble, France
†deceased, 8 July 2023

**Correspondence:** Yumeng Chen (yumeng.chen@reading.ac.uk)

**Abstract.** In this study, we investigate the fully multivariate state and parameter estimation through idealised simulations of a dynamic-only model that uses the novel Maxwell-Elasto-Brittle (MEB) sea ice rheology and in which we estimate not only the sea ice concentration, thickness and velocity, but also its level of damage, internal stress and cohesion. Specifically, we estimate the air drag coefficient and the so-called damage parameter of the MEB model. Mimicking the realistic observation
network with different combinations of observations, we demonstrate that various issues can potentially arise in a complex sea ice model especially in instances for which the external forcing dominates the model forecast error growth. Even though further investigation will be needed using an operational (a coupled dynamics-thermodynamics) sea ice model, we show that, with the current observation network, it is possible to improve both the observed and unobserved model state forecast and parameters accuracy.

**1   Introduction**

An accurate representation of the state of sea ice in the Arctic is important for making both short-term, seasonal and long-term climate predictions. Recent observations show that its extent is in decline, and, in particular, that it is shifting from a multi-year ice type to "younger" first-year ice (Meier, 2017). This ongoing shift induces a larger year-to-year and inter-annual variability in the sea ice extent, that makes the short-term and seasonal Arctic sea ice forecasting even more challenging, albeit its crucial
relevance for shipping routes and fisheries (Mioduszewski et al., 2019). The Arctic sea ice is also a major player of the climate systems via its feedback to the Earth surface albedo, ocean and atmosphere global circulations. Accurate sea ice simulations are therefore essential for better climate projections (Bertino and Holland, 2017).

As in other areas of environmental predictions, errors in sea ice models can be attributed to either errors in the initial conditions or in the model representation of physical processes. Due to the low degree of internal instabilities of the sea-ice



evolution on long space and time scales (> seasonal timescales), model error is the main source of prediction uncertainty and it is especially detrimental to long-term forecasts needed for climate simulations. Numerical models incorporate parametrisations to represent physical processes that are not explicitly described or resolved in the model. The parameters involved can be estimated in different ways. One way is to base the parameter on values obtained from laboratory experiments in idealised environments, but has the limitation that the selected values may not scale up for complex realistic simulations. Alternatively,

one can select a value from a range of possible candidates based on which candidate produces the best fit between model and observations (Dansereau, 2011; Miller et al., 2006; Heorton et al., 2019). Nevertheless, as the number of parameters to be tuned increases, the latter approach becomes computationally infeasible. In that case, an optimised version of the aforementioned process called "data assimilation" must be pursued.

Data assimilation (DA) combines observations with the model forecast to provide the most-likely estimate for the true

state/parameter of the system (see e.g., Carrassi et al., 2018; Evensen et al., 2022; Park and Zupanski, 2022). It has been proved to be essential for sea ice forecasting (Sakov et al., 2012a; Lea et al., 2015; Zuo et al., 2019). Data assimilation can account for both the forecast and observation errors in the state/parameter estimation. To this extent, error statistics need to be specified. In ensemble DA, the forecast error statistics are obtained from an ensemble of model runs (Evensen, 2003).

Previous studies have shown that assimilating sea ice concentration (SIC) can reduce errors in sea ice thickness (SIT) (Mas-

sonnet et al., 2015) and that, in coupled sea-ice and ocean models, sea ice observations help initializing ocean fields such as the ocean and wind forcing (Toyoda et al., 2016). Model parameters can be made part of the state and when it is assumed that they are affected by errors, DA can be used to estimate them. An example of such an approach in sea-ice models can be found in Massonnet et al. (2014) who used the ensemble Kalman filter (EnKF, Evensen, 2003) to estimate the air drag coefficients, ocean drag coefficients and the sea ice strength parameter by assimilating sea ice drift data.

Sea ice models include multiple variables. However, only three of these are usually observed and assimilated: SIC, SIT and sea ice drift. These observations are mostly obtained from satellites. In-situ observations are usually only used for evaluation purposes (Jakobson et al., 2012). The satellite observations of SIC show good spatial and temporal coverage with relatively low uncertainty. Assimilation of SIC shows tremendous benefits for sea ice forecasting (Lisæter et al., 2003; Stark et al., 2007). In the past decades, though less accurate than the SIC, SIT has started to be assimilated, with further improvements to the

forecasts (Xie et al., 2018). The assimilation of sea ice drift has been comparatively less successful and motivated further developments of sea ice rheological models (Sakov et al., 2012a). These studies altogether show the important role of DA in sea ice prediction.

In this study, we explore the capability of ensemble DA to estimate both the state and key sea ice parameters in a model endowed with an MEB rheology. This rheology is adopted by the neXt-generation Sea Ice Model (neXtSIM) which runs

operationally on a Lagrangian grid that uses dynamical remeshing (Rampal et al., 2016). Due to the Lagrangian framework, the advection process does not require explicit calculations, and the highly multi-scale sea ice features, especially localised features such as the so-called Linear Kinematic Features (LKFs), which concentrates sea ice fracturing and high deformations rates, can be easily preserved (Bouillon and Rampal, 2015). One downside of this framework however is that, with changing number of model grid points in time, it poses challenges for ensemble DA. Consequently, DA approaches have been specifically



designed (e.g., Aydoğdu et al., 2019; Sampson et al., 2021) and implemented in the context of neXtSIM by Cheng et al. (2023). Another drawback of using a Lagrangian framework in a sea ice model is that coupling to existing ocean and atmospheric model components, which are virtually all Eulerian, requires the use of a coupler on a fixed exchange grid (Boutin et al., 2023).

As such a coupling is essential for climate simulations, a new version of neXtSIM based on an Eulerian framework is being developed under the Scale-Aware Sea Ice Project (SASIP; https://sasip-climate.github.io/). It is in the light of these

developments that this study is inscribed. In particular, the dynamics-only sea ice MEB model of Dansereau et al. (2016, 2017) has been designed following a Eulerian approach and a Discontinuous-Galerkin treatment of advection. With the aid of this dynamics-only sea ice MEB model, we study in detail the capabilities, limitations and adaptations of ensemble DA to infer state and model parameters based on synthetic observations of the Arctic sea ice.

We intentionally use a dynamics-only model whereby thermodynamics processes are missing. Indeed, such model is already

complex enough and on short (daily and weekly) times scales, sufficient to focus on how the mechanical/dynamical processes lead to the emergence of complex, potentially nonlinear, relations between model states, its parameters and the observable quantities. Those are the relations the DA has to rely on. Thus, although the model does not capture all of the processes at play in sea ice, we shall see how our experiments reveal a number of complex interactions. On the other hand, the use of a simpler idealised model allows us to conduct a fully multivariate estimate where we infer all model fields with only a handful

of available observations, a realistic situation that has not yet studied extensively in sea-ice DA.

The paper is organised as follows. In Sect. 2 we introduce the iterative ensemble Kalman filter (IEnKF), state-of-the-art DA approach used in this study. Then, we describe the model and its configurations in Sect. 3. The ensemble DA setup and twin experiments are given in Sect. 4. Results for both multivariate state estimation in "perfect model" setup and parameter estimation in a biased model are reported in Sect. 5. In Sect. 6 we discuss some of the critical aspects of ensemble DA in this

context together with how to address them in future works. We finally summarise our findings in Sect. 7.

## 2   Data assimilation: The iterative ensemble Kalman filter

The iterative ensemble Kalman filter (IEnKF, Sakov et al., 2012b; Bocquet and Sakov, 2012, 2014) is a variant of the Kalman filter (KF). Like the KF, it is constructed based on the Bayes theorem under the assumption that the prior (forecast), evidence (observation) and posterior (analysis) follow a Gaussian distribution. It has successfully been applied to joint state and model

parameter estimation problems (Bocquet and Sakov, 2013; Haussaire and Bocquet, 2016; Bocquet et al., 2021). In ensemble DA methods, the linear model assumption of the KF is relaxed and the prior distribution is estimated from a finite ensemble of model forecasts. A relevant feature of the IEnKF is that it solves for the analysis via a nonlinear optimisation aimed at maximising the aposterior probability distribution. This key aspect makes it worth investigating its performance in the context of predicting Arctic sea ice, which is characterised by strong and complex nonlinear relations as well as weak non-linearity in

the observations. The IEnKF is the filter version of the more general iterative ensemble Kalman smoother (Bocquet and Sakov, 2014) and it is conceptually a generalisation of the maximum likelihood ensemble filter by Zupanski (2005).



In an ensemble DA system consisted of $N_e$ ensemble members with $N$-dimensional state vector, the ensemble mean of the posterior analysis $\overline{\mathbf{x}}^a \in \mathbb{R}^{\mathbb{N}}$ is given by $\overline{\mathbf{x}}^a = \overline{\mathbf{x}}^f + \mathbf{A}\mathbf{w}_{min}$ with $\overline{\mathbf{x}}^f \in \mathbb{R}^N$ being the apriori ensemble mean, $\mathbf{A} \in \mathbb{R}^{N \times N_e}$ the matrix of ensemble anomalies with $N$ rows and $N_e$ columns obtained by removing the ensemble mean from the full ensemble matrix, and $\mathbf{w}_{min} \in \mathbb{R}^{N_e}$ being the minimum of $\mathbf{w}$ obtained from the cost function as

$$\mathbf{w}_{min} := \underset{\mathbf{w}}{\operatorname{argmin}} \mathcal{J}(\mathbf{w}), \tag{1a}$$

with

$$\mathcal{J}(\mathbf{w}) = \frac{1}{2}\left(\mathbf{y} - \mathcal{H}(\overline{\mathbf{x}}^f + \mathbf{A}\mathbf{w})\right)^{\mathrm{T}}\mathbf{R}^{-1}\left(\mathbf{y} - \mathcal{H}(\overline{\mathbf{x}}^f + \mathbf{A}\mathbf{w})\right) + \frac{1}{2}(N-1)\mathbf{w}^{\mathrm{T}}\mathbf{w}, \tag{1b}$$

where $\mathbf{y} \in \mathbb{R}^{N_o}$ contains $N_o$ observations whose error covariance is specified by $\mathbf{R} \in \mathbb{R}^{N_o \times N_o}$ and $\mathcal{H}$ is the observation operator. Formulating the problem as in Eq. (1) is equivalent to an ensemble variational method. Note that extensive reviews of these methods can be find in the chapter 7 of Asch et al. (2016), Bannister (2017), or Sect. 4 in Carrassi et al. (2018). In our implementation, the cost function in Eq. (1) is minimised using a Gauss-Newton method. The stopping criterion in this study is to have a maximum 40 number of iterations when performing state-estimate only (Sect. 5.1 and 5.2) with the additional constraint of $||\mathbf{w}^k - \mathbf{w}^{k-1}|| < 10^{-3}$ when parameters are also estimated (Sect. 5.3-Sect. 5.5). The posterior analysis error covariance is approximated, via the ensemble, by the inverse of the Hessian of $\mathcal{J}$.

The forecast error covariance is approximated with the ensemble anomaly matrix, $\mathbf{A}$, by $\frac{1}{N-1}\mathbf{A}\mathbf{A}^{\mathrm{T}}$. Given that the state vector contains both observed and unobserved model fields, the forecast error covariance matrix contains the (ensemble-based) cross-covariance between these fields that allow for propagating the data content to all model fields, including those that are not directly observed. As we shall clarify later, we use a fully multivariate augmented DA where the state vector contains all model fields and model parameters that we will be estimated (see for instance Ruiz et al., 2013b). The analysis of the unobserved fields and parameters depends directly on the cross-correlations between these fields and the observations.

## 3 Model setup

In this study, we use a dynamics-only sea ice model in an idealised setup. In this section, we provide details about the model setup used in our numerical experiments. This includes the model equations, its parameters and the choice of external forcing fields. The setup described in this section is used as modelled "truth" in following DA experiments.

### 3.1 The dynamics-only sea ice model

The dynamics-only sea ice model uses the MEB rheology proposed by Dansereau et al. (2016). The model itself and its numerical implementation, based on an Eulerian, discontinuous Galerkin approach, is presented in Dansereau et al. (2017). In the MEB model, sea ice is treated as a viscous-elasto-brittle material. This rheology allows for representing the ice cover as a brittle solid where it is relatively undamaged (i.e., unfractured) and highly concentrated relative to ice-free waters, and as an elastic-viscous fluid where it is intensively fractured and low in concentration. By associating sea-ice to an elastic solid



rather than a highly viscous fluid and by incorporating a variable (the level of damage, $d$) to represent its degree of fracturing at the sub-grid scale, this model differs significantly from the widely-employed (elastic-)viscous-plastic rheologies (Hunke et al., 2010).

The dynamics-only MEB sea ice model describes the evolution of 9 model fields: sea ice concentration (SIC) $A$, thickness (SIT) $h$, velocity (SIV) $\mathbf{u} = (u_x, u_y) = (u, v)$, level of damage $d$, cohesion $C$, and internal stress $\sigma = \begin{pmatrix} \sigma_{xx} & \sigma_{xy} \\ \sigma_{yx} & \sigma_{yy} \end{pmatrix}$, where $\sigma_{yx} = \sigma_{xy}$ (see Dansereau et al., 2017).

In the MEB rheology, the evolution of the SIV, level of damage, and stress describes the kinematics of the sea ice. To obtain a physically plausible solution, a *stress-velocity-damage constraint* that respects the MEB constitutive equation (which relates

SIV to $\sigma$) must be satisfied. Other model fields also enter the momentum, constitutive, damage evolution and mass conservation equations, but these are just advected by the SIV. Here, we highlight the momentum and stress equations of the sea ice model. These equations will be relevant in parameter estimation experiments. The momentum equation is given as:

$$\rho h \frac{D\mathbf{u}}{Dt} = \nabla \cdot (h\sigma) + \rho_a C_a A |\mathbf{u_a}| \mathbf{u_a}, \tag{2}$$

where $\rho$ and $\rho_a$ are the sea ice and air density respectively, $\mathbf{u_a}$ is the wind field, $\frac{D}{Dt}$ is the material derivative, and $C_a$ is the air

drag coefficient. The stress equation is:

$$\lambda^0 d^{\alpha-1} \left[ \frac{\partial \sigma}{\partial t} + (\mathbf{u} \cdot \nabla)\sigma + \beta_a(\nabla\mathbf{u}, \sigma) \right] + \lceil 1 - d \rceil \sigma = \eta^0 d'^\alpha \exp\left[-c^*(1-A)\right] \mathbf{K} : \mathcal{D}(\mathbf{u}), \tag{3}$$

where $\lambda^0$ is the undamaged relaxation time; $\beta_a$ is a function that accounts for the effects of rotation and deformation; $\eta^0$ is the undamaged apparent viscosity; $d'^\alpha = (1 - \frac{\eta_{min}}{\eta_0})d^\alpha + \frac{\eta_{min}}{\eta_0}$ with $\eta_{min}$ being the minimum apparent viscosity and $\alpha$ being a damage-related parameter; $\mathbf{K}$ is a stiffness tensor and $\mathcal{D}()$ is the symmetric part of the velocity gradient.

The model equations are discretised on an unstructured triangular grid using a Finite Element, Discontinuous Galerkin method where the sea ice velocities are defined on triangular vertices (degree of polynomial approximations of 1) and all other model fields are defined on the face of the triangular element (degree of polynomial approximations of 0, or constant by element, see Dansereau et al., 2017). In particular, because the constitutive, momentum and level of damage evolution are coupled, SIV, level of damage and sea ice stress are solved using a semi-implicit, iterative method. Besides these dynamical

properties, the sea ice model used in this study does not include thermodynamics processes. As such, it is a short time-scales proxy of the dynamic-thermodynamic sea ice model currently under construction in the SASIP project, neXtSIM$_{\mathrm{DG}}$. However, this future model builds on the MEB rheology and Discontinuous Galerkin-based numerical scheme. Therefore, the present model is a reliable surrogate of the dynamical components and of the numerical core of neXtSIM$_{\mathrm{DG}}$.

The evolution of the sea ice model is controlled by multiple model parameters, which define the physical properties of the sea

ice and thereby intrinsically affects the representation of sea ice in the model. These parameters (cf. Tab. A1) follow the choice in Dansereau et al. (2017) with exceptions for the spatial and temporal resolutions and minimum cohesion. In our experiments, the model runs at the spatial resolution of around $15\,\mathrm{km}$ and a time step of 30 seconds. This ensures numerical stability while sufficiently resolving the propagation of damage, the fastest process represented in the model. The model is solved on a squared





**Table 1.** The initial and boundary condition of the experimental setup in the dynamics-only sea ice model. Here $r = \sqrt{x^2 + y^2}$ with $(x, y)$ being the coordinate of the grid points and $r_0 = 50$ km. The boundary condition at $y = -L = L$ is that of the fields transported into the model domain.

| Initial condition | Boundary condition | | Forcing | |
|---|---|---|---|---|
| | $x = -L = L$ | $y = -L = L$ | ocean | atmosphere |
| $A = 1$ | | $A = 1$ | | |
| $d = 0$ | | $d = 0$ | | random quasi-periodic storm-like |
| $\mathbf{u} = 0 \, \mathrm{m} \cdot \mathrm{s}^{-1}$ | $\mathbf{u} = 0 \, \mathrm{m} \cdot \mathrm{s}^{-1}$ | $\sigma(t) \cdot \mathbf{n} = 0 \, \mathrm{Pa}$ | at rest | |
| $\sigma = 0 \, \mathrm{Pa}$ | | $\sigma = 0 \, \mathrm{Pa}$ | | |
| $h = \max(1, 1 + \cos(\frac{\pi}{2} r/r_0)) \, \mathrm{m}$ | | $h = 1 \, \mathrm{m}$ | | wind field occurrence |
| $C \sim \mathcal{U}(5000, 10000) \, \mathrm{Pa}$ | | $C \sim \mathcal{U}(5000, 10000) \, \mathrm{Pa}$ | | |

model domain of $[-L, L] \times [-L, L]$ with $x$- and $y$-axis aligned along the perpendicular sides of the square with $L = 100$ km.

This gives us 512 elements and 285 nodes. To enhance internal variability (and thus maintain ensemble spread in the IEnKF), the minimum sea ice cohesion, which sets the resistance of the sea ice cover from fracturing, is lowered to $5,000 \, \mathrm{Pa}$ from $8,000 \, \mathrm{Pa}$ used so far in neXtSIM (Rampal et al., 2016).

In this study, DA's ability to estimate two model parameters will be investigated: the air drag coefficient, $C_a$, in Eq. (2) and damage parameter, $\alpha$, in Eq. (3). $C_a$ enters the model in the momentum equation (cf. Eq. (2)) modulating the influence of

the wind fields on the SIV. This parameter corresponds to the effect of external forcing on the sea ice model. $\alpha$ controls the swiftness of the transition between the elastic-brittle regime at low level of damage and the viscous regime at high damage. The mechanical behaviour of an elastic-brittle solid is less predictable in nature than that of a viscous fluid (Weiss and Dansereau, 2017). This means that erroneous $\alpha$ changes the internal property of sea ice and influences the sea ice predictability.

The initial and boundary conditions of the model are shown in Tab. 1. The simulations start with a domain covered by undam-

aged sea ice at rest with a random cohesion field. The value of the cohesion field for each element is sampled from a uniform distribution between $5,000$ and $10,000 \, \mathrm{Pa}$. As shown in Fig. 1e, the initial SIT ($h$) is a "blob" defined by a cosine function given in Tab. 1, which represents the naturally inhomogeneous distribution of the SIT in space. We use no-slip boundary conditions at $x = -L = L$ and Neumann boundary conditions at $y = -L = L$ where 1 m-thick undamaged sea ice is transported into the domain based on the SIV. To avoid an influx of sea ice with uniform cohesion field from the domain boundaries, the

cohesion field of the inflowing ice is randomly sampled from a uniform distribution between $5,000$ and $10,000 \, \mathrm{Pa}$.

## 3.2 The external wind field

In the model setup, following Dansereau et al. (2017), the ocean is assumed to be at rest and the sea ice variability relies on the wind forcing. We design an external forcing in the form of a prescribed storm-like wind drag that mimics the wind field



**Table 2.** The parameters of the storm served as the truth of the wind field.

| period (days) | strength (m·s$^{-1}$) | | | initial centre (km) | | travel speed (m·s$^{-1}$) |
|---|---|---|---|---|---|---|
| | rotational | divergent | background | x | y | |
| $\mathcal{U}(2,5)$ | 22 | 0.1 | 2 | $\mathcal{U}(-25,25)$ | $\mathcal{U}(-50,20)$ | 0.25 |

encountered in operational sea-ice DA. Consistent with reality, the wind is therefore a major source of variability in the present
simulations (Guemas et al., 2016; Rabatel et al., 2018).

The storm-like wind field is inspired by the test cases for linear advection schemes in Lauritzen et al. (2012) generated by
analytical formulae as outlined in Appendix B. We deem this forcing an adequate representation of the complex horizontal
atmospheric flow. The wind field is updated at each model computational time step such that the sea ice is always driven by
the up-to-date wind field similar to the case where the sea ice model is coupled to an atmospheric model.

The cyclone, shown in Fig. 1a, covers around a quarter of the domain in space. The wind field is superposed on a background
flow ($2\,\mathrm{m}\cdot\mathrm{s}^{-1}$) from the bottom to the top of the domain. The centre of the storm moves from $y=0$ to $y=L$ with a speed of
$0.25\,\mathrm{m}\cdot\mathrm{s}^{-1}$ which is slower than the background flow. The parameters of the storm can be found in Tab. 2. The formulae for
the wind field also allow for a fine control of the generation and dissipation of the cyclone. Therefore, as shown in Fig. 1b, the
storms (appearing as peaks in the time-series) are not persistent but have time-evolving features. Following the dissipation of
a cyclone, a "peaceful" period with only the background wind is used. Hence, during our 90-day experiment period, there are
12 storm occurrences. Moreover, like the duration, the strength, the initial centre position, and the travel speed of the storm is
specified by the analytical formula with storm duration and initial position sampled from a uniform distribution as shown in
Tab. 2. With the given random initial position of the storm, the variability of the wind field is confined in a limited region of the
domain as shown in Fig. 1c and d. Figure 1e-f shows that the variability of the wind field leads to a variability of the modelled
185    sea ice mainly in the upper right region of the domain as the sea ice is mostly damaged in these regions.

## 4    Data assimilation setup

The operational sea ice DA-only corrects the model state without dealing with model errors explicitly. It is however impos-
sible to dispose of a perfect model. Model errors originate from diverse sources, such as numerical discretization, the lack of
sufficient resolution as well as errors in the model parameters. We focus here on the parametric errors, as these are particularly
190    problematic and dominate over the initial conditions errors in long-term forecasts.

As introduced in Sect. 2, the IEnKF can infer unobserved model fields thanks to its ensemble-based cross-correlations
with the observed fields. This feature can also be exploited for parameter estimations. In the parameter estimation, we use
the augmented approach in which the model parameters are included as part of the state vector. For example, in the case of
estimating the air drag coefficient $C_a$, the state vector is $\mathbf{x} = (h, u, v, \ldots, C_a)^{\mathrm{T}}$. Thus their inference is ultimately related to the





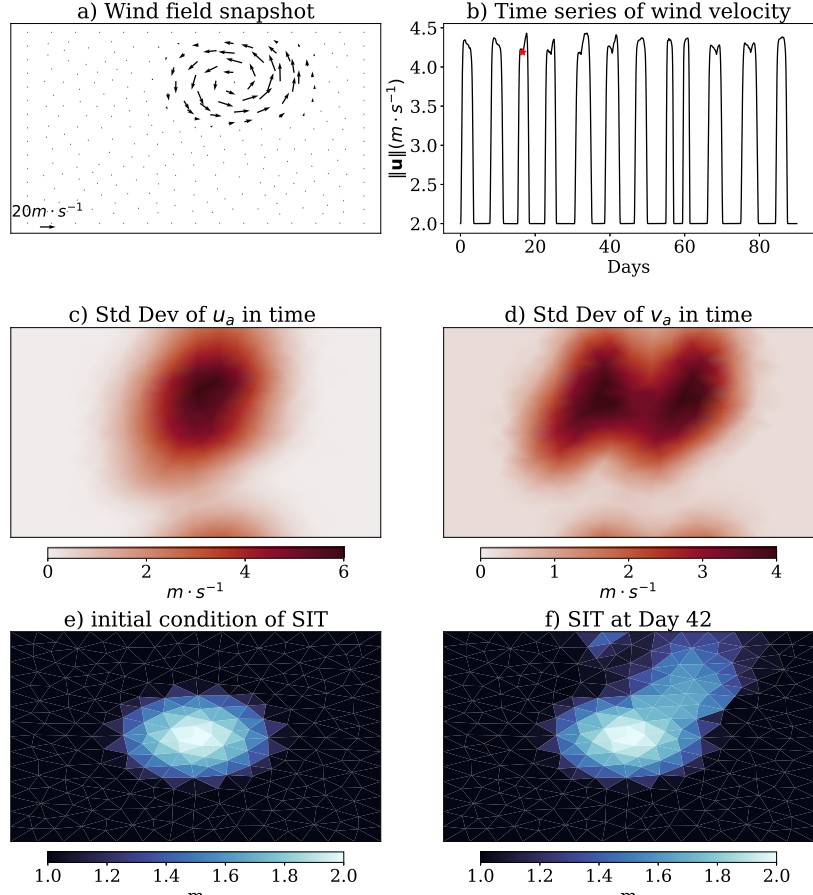

**Figure 1.** An illustration of the wind field and the model response in SIT. a) A snapshot of the wind field at an arbitrary time; b) time series of the spatial-averaged wind velocity; c) Standard deviation of the $u$-component and d) the $v$-component of the wind in time over 90 days; e) initial condition of the SIT; f) SIT after 42 days of simulation.

capability of the IEnKF to properly describe the correlation within this new, augmented virtual state composed by the physical variables and the parameters.

The simultaneous state and parameter estimation is more challenging than state estimation alone since changes in the parameters have a direct impact, and can substantially modify the model's dynamical properties. In the worst scenario, inappropriate parameter values can push the model through a bifurcation (a tipping point in the case of non-autonomous systems) leading to a qualitatively different model behaviour than those in the data. Moreover, different parameters can have aliased uncertainties in the observed model fields, implying that not all of the parameters can be uniquely identified.

These challenges motivate the investigation of state and parameter estimation in the idealised dynamics-only sea ice model. In this study, a series of twin experiments are conducted whereby a model run is taken to represent the "truth". Synthetic observations are generated from the "truth" following a specified observation error distribution.



**Table 3.** The experiment setup used in four different scenarios. These experiments use the same truth where $\mathbf{u}_a$ is the wind forcing, $\mathbf{u}(t = 0)$ is the initial condition of the SIV.

| | model error | inference vector | perturbations | background | | inflation | | localisation (km) | | |
|---|---|---|---|---|---|---|---|---|---|---|
| | | | | $C_a$ | $\alpha$ | state | $C_a/\alpha$ | SIC | SIT | SIV |
| 1. | None | state | $\mathbf{u}, \mathbf{u}_a$ | $1.5 \times 10^{-3}$ | 4 | IEnKF-N | | 5 | 200 & 30 | 30 |
| 2. | $C_a$ | state, $C_a$ | $\mathbf{u}, \mathbf{u}_a, C_a$ | $2.5 \times 10^{-3}$ | 4 | 1.02 | Eq. (7)* | 5 | 30 | 30 |
| 3. | $\alpha$ | state, $\alpha$ | $\mathbf{u}, \mathbf{u}_a, \alpha$ | $1.5 \times 10^{-3}$ | 6.5 | 1.02 | Eq. (7)* | 5 | 30 | 30 |
| 4. | $C_a$ and $\alpha$ | state, $C_a, \alpha$ | $\mathbf{u}, \mathbf{u}_a, C_a, \alpha$ | $2.5 \times 10^{-3}$ | 6.5 | 1.02 | Eq. (7)* | 5 | 30 | 30 |

*A time-dependent inflation factor.

To assess DA's ability, four different scenarios are explored: 1) a perfect model where the parameters are equal to their "true" values; 2) a model with biased air drag coefficient, $C_a$; 3) a model with biased damage parameter, $\alpha$; 4) a model with biased $C_a$ and $\alpha$. A summary of the setup of each experiment is given in Tab. 3.

## 4.1 Ensemble generation

The IEnKF belongs to the category of the ensemble-based DA methods. As such it relies on an ensemble of model trajectories to approximate the forecast uncertainty. The ensemble spread represents the error in the estimate of the model state and parameters. We will explore four different scenarios, whereby, as we shall clarify later, we will employ each time a different strategy to generate the ensemble. In particular, in the cases with parametric error, each member of the ensemble will be given a different set of model parameters.

Nevertheless, in all of the four scenarios, we will perturb both the wind field (i.e. an external forcing), the initial condition of the sea ice velocity, and the cohesion flux at model boundaries. The external atmospheric wind forcing is a major source of forecast uncertainty in sea ice models. For example, Rabatel et al. (2018) and Cheng et al. (2020) studied the sensitivity of neXtSIM to the wind forcing and sea ice cohesion. In this study, we generate synthetic perturbed wind fields around the "true wind" defined in Sect. 3.2 and use them to form an ensemble. The duration, strength, initial position, and the travel speed of each cyclone are perturbed with noises sampled from Gaussian distributions (see Tab. 4 for details). In addition to these perturbations, we also introduce a random walk for the centre of the cyclone in the zonal direction as part of the ensemble perturbation. This is generated as a red noise at each time step according to

$$\Delta x_{i+1} = e^{-\Delta t/\tau} \Delta x_i + \epsilon, \tag{4}$$

where $\Delta x_i$ is the distance travelled zonally at $i$th time step, $\tau = 60$ is a time decorrelation factor, $\epsilon$ is a noise sampled from $\mathcal{N}(0, 4.44 \times 10^{-5})$ m. This red noise is not applied to the "true wind". To avoid the cyclone traveling outside of our model domain, we resample the noise if the centre of the cyclone goes outside of the region between $-40$ km and $50$ km in the $x$-coordinate. The same check and re-sampling is applied to the perturbations for the initial centre of the cyclone.





**Table 4.** Gaussian distributions of wind fields, SIV and parameter perturbations. The mean of the wind fields is given in Tab. 2.

| | wind field | | | | | SIV (m·s$^{-1}$) | $C_a$ | $\alpha$ |
|---|---|---|---|---|---|---|---|---|
| period (days) | strength (m·s$^{-1}$) | | initial centre (km) | | travel speed (m·s$^{-1}$) | | | |
| | rotational | background | x | y | | | | |
| $\mathcal{N}(0,0.5)$ | $\mathcal{N}(0,0.5)$ | $\mathcal{N}(0,\sqrt{0.05})$ | $\mathcal{N}(0,1.5)$ | $\mathcal{N}(0,1.5)$ | $\mathcal{N}(0,0.0045)$ | $\mathcal{N}(0,0.05)$ | $\mathcal{N}(0,5\times10^{-4})$ | $\mathcal{N}(0,1.5)$ |

**Table 5.** The time- and space-averaged ensemble standard deviation of each free run ensemble over 90 days.

| Scenario | $u$ (m·s$^{-1}$) | $v$ (m·s$^{-1}$) | $A$ | $h$ (m) | $d$ | $C$ (Pa) | $\sigma_{xx}$ (Pa) | $\sigma_{xy}$ (Pa) | $\sigma_{yy}$ (Pa) |
|---|---|---|---|---|---|---|---|---|---|
| 1. | $7.52\times10^{-3}$ | 0.011 | 0.072 | 0.037 | 0.215 | 153.128 | 2584.221 | 1475.218 | 1701.683 |
| 2. | 0.030 | 0.031 | 0.153 | 0.089 | 0.240 | 289.626 | 3089.592 | 1508.822 | 1925.960 |
| 3. | $7.42\times10^{-3}$ | 0.011 | 0.074 | 0.038 | 0.185 | 166.199 | 2657.060 | 1502.896 | 1754.829 |
| 4. | 0.033 | 0.034 | 0.149 | 0.093 | 0.210 | 295.344 | 3139.044 | 1517.597 | 1951.430 |

Albeit marginal with respect to variability due to the external forcing, the internal model variability caused by non-linearities is another source of forecast error. This is accounted for in our ensemble by perturbing the SIV initial condition, in addition to perturbing the external atmospheric wind field. SIV perturbations are sampled from the Gaussian distribution, $\mathcal{N}(0,0.05)$.

Furthermore, as discussed in Sect. 3.1, the boundary condition of the random cohesion influx, which differs for each ensemble member, adds another source of uncertainty in our experiments but as the sea ice drifts slowly and does not travel long distances across the domain, the impact of the cohesion perturbation is limited.

In the experiments with parametric error in either or both $C_a$ and $\alpha$ (Exp 2–4 in Tab. 3) the prior covariances of the model parameters need to be specified. As shown in Tab. 4, the parameter values used by the ensemble members are sampled from

zero-mean Gaussian distributions with standard deviations set to be around $33\%$ of the true parameter value. Given that both $C_a$ and $\alpha$ are bounded from below due to physical constraints, the sampled values of $C_a$ and $\alpha$ are ensured to be greater than $10^{-5}$ and 2 by rejecting outliers.

## 4.2 Synthetic observations

Synthetic observations in our twin experiments are generated by sampling from the "truth" with the aim to mimic how obser-

vations of the Arctic sea-ice are collected operationally, e.g. their spatio-temporal density.

Satellite observations of SIC, SIT and sea ice drift are used in sea ice forecasting and reanalysis systems (Sakov et al., 2012a). Most operational systems assimilate gridded products. However, some recent studies show the possibility of directly assimilating along-track instead of gridded data for SIT (Fiedler et al., 2022), which could allow for a more frequent SIT assimilation than the 7-day averaged gridded products (Ricker et al., 2017) because gridded data requires time for collection and processing.





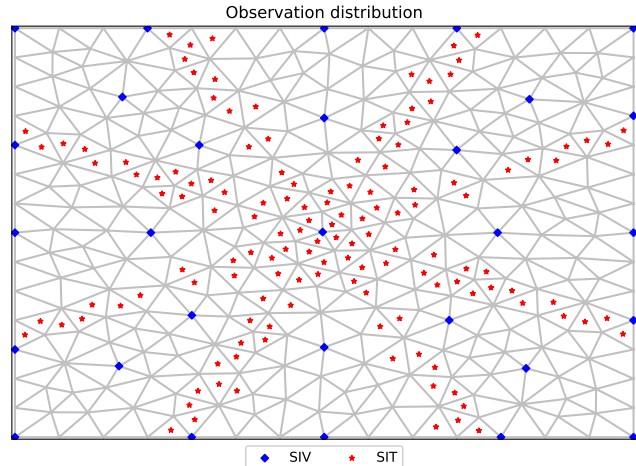

**Figure 2.** Observation distribution of SIT and SIV with the background triangles being the model grid. The observation distribution of SIC is not shown because it covers the entire domain.

Hence, a star-shaped spatial distribution mimicking the along-track SIT observations is used here. For simplicity, we assume that the same star-shaped spatial distribution is available daily due to the time required for data collection from polar satellite. Note that this treatment neglects the temporal variability of the satellite tracks. Following the protocol of gridded data, we generate synthetic observations of SIC and SIV quasi uniformly across the model domain. Nowadays SIC satellite observations reach a resolution as high as $10$ km. Considering that our model has a spatial resolution of around $15$ km, we synthetically ob-

serve every grid point for SIC in our experiments. SIV data are sparser, with a spatial resolution of around $50$ km. The position of the along-track SIT data is parametrised by four lines on the domain satisfying the condition $|y_i + cx_i| \leq r$ where the $(x_i, y_i)$ is the spatial coordinate of the grid points, and the pair $(c, r) \in \{(2.2, 16 \text{ km}), (-2.2, 16 \text{ km}), (0.5, 8 \text{ km}), (-0.5, 8 \text{ km})\}$. A graphic distribution of SIC and SIV can be found in Fig. 2.

  The synthetic observations are generated by sampling from the "truth" and adding an observational error drawn from Gaus-

sian distributions. The observation error variances of these Gaussians follow those of realistic DA systems. For SIC, we adopt the observation error standard deviation from Sakov et al. (2012a) and Cheng et al. (2023):

$$\sigma_A[m] = \sqrt{0.01 + (0.5 - |0.5 - A|)^2}. \tag{5}$$

The observation error of the along-track SIT follows the formula for the measurement uncertainty for CryoSat-2 used by Fiedler et al. (2022),

$$\sigma_h = \begin{cases} 8 & h < 0.7 \text{ m} \\ 2\left(1 - \frac{7e}{0.3 - h^e}\right)\frac{h}{100} & 0.7 \leq h < 3 \text{ m} \\ \left(5(h - 3) + \left(1 - \frac{7e}{0.3 - 3^e}\right)\right)\frac{h}{100} & h > 3 \text{ m}. \end{cases} \tag{6}$$



This equation parameterises the uncertainty of the observations based on the measured value of the SIT. The use of $\sigma_h = 8$ m for $h < 0.7$ m effectively eliminates SIT observations of thin ice from the DA reflecting that SIT of thin ice from satellite altimetry is notoriously untrustworthy. The error in Eq. (6) does not account for the representation errors. To account for it, we have therefore added an extra factor of 2.

For the sake of simplicity, synthetic SIV observations are used as a substitute to the sea ice drift data. There is no explicit equation for the observational error variance of the SIV, nor prototypical examples, due to the lack of investigation of SIV uncertainties. Hence, for each observation grid point, we use either $80\%$ of the standard deviation of a single 90-day model trajectory (in our case, the truth) of the observation variance, or $8 \times 10^{-4}$ m $\cdot$ s$^{-1}$ ($\sim 0.21$ km/3 days), whichever is greater. This leads to a maximum observational error around $12$ km/3 days ($13$ km/3 days) for $u(v)$-component and insures that SIV observations will contribute to the DA correction. However, the value for the standard deviation is smaller than the $14$ km/3 days value used in Sakov et al. (2012a): one of the few cases of operational reanalyses using SIV we are aware of. Since Sakov et al. (2012a) showed only a faint impact of SIV assimilation, a reduction of the observation error is expected to take better advantage of the SIV data.

### 4.3 Inflation and localisation

#### 4.3.1 Inflation

The application of ensemble DA in geosciences is plagued by sampling errors arisen by the impossibility to use sufficiently large ensembles. The huge size of realistic numerical models of geofluids and the computational constraints imply that the number of affordable ensemble size is much smaller than the state vector's dimension, $N_e \ll N$. Inflation and localisation are the two main approaches to alleviate sampling errors (and to some extent, model errors, as discussed in Scheffler et al. (2022); Grudzien et al. (2018)). Following Cheng et al. (2023), an ensemble size of $N_e = 40$ is used for the IEnKF. The ensemble size is thus orders of magnitude smaller than the size of the state vector which is $\mathcal{O}(10^3)$.

We use different inflation strategies in our four experimental scenarios (cf. Tab. 3). In *scenario 1*, when we only infer the model state vector, we use the adaptive inflation method. The IEnKF-N is an extension of IEnKF that includes an adaptive inflation method designed to counteract the sampling error by keeping a safe ensemble spread (Bocquet and Sakov, 2012). Although the IEnKF-N has a slightly larger computational cost, it spares us from the very costly offline tuning of the inflation factor (a procedure that should ideally be repeated for each experimental setup).

However, as we will further discuss in Sect. 6, the cross-correlation between observed (physical) quantities and the parameter in the ensemble can be undesirably modified by the adaptive inflation procedure of IEnKF-N. Therefore, we use the standard IEnKF without adaptive inflation whenever simultaneously inferring state and model parameters (*scenarios 2-4*). In these experiments, we separate the inflation for the physical state variable and that for the parameter.

In the adjoint state and parameter estimations, we use a fixed multiplicative inflation factor of $1.02$ applied exclusively to the physical variables of the model (i.e. it does not affect the ensemble-based estimate of the error variance in the parameter). We make several tests to identify the value of $1.02$ as satisfactory.



Given that the evolution of the parameter is governed by the persistence model, the estimated parametric error stays constant
between successive analyses. On the other hand, at the analysis times, the parametric error is bounded to stay unaltered or
to be reduced. The overall result of these two effects is that the filter progressively gains confidence in its estimate of the
parameter and reaches convergence around a value, no matter whether or not this value is actually correct (Ruiz et al., 2013a).
To counteract this effect we apply an ad-hoc adaptive multiplicative inflation factor $\lambda$ such that the forecast ensemble spread of
the model parameter is always above a chosen lower bound, $\sigma_{\mathrm{std}}^t$. Before each analysis step, the inflation is obtained according
to

$$\lambda = \frac{\max(\sigma_{\mathrm{std}}^t, \sigma_{\mathrm{std}}^f)}{\sigma_{\mathrm{std}}^f}. \tag{7}$$

The lower bound is defined as $\sigma_{\mathrm{std}}^t = p\sigma_{\mathrm{std}}^0$, with $\sigma_{\mathrm{std}}^0$ being the standard deviation of the initial uncertainty of the model
parameter as given in Tab. 4, while $p = 0.4$ is a scaling factor; in Eq. (7), $\sigma_{\mathrm{std}}^f$ is the forecast spread at the current DA step. With
this time-dependent inflation, the forecast ensemble spread in the model parameter is always larger than $40\%$ of the initially
specified uncertainty.

### 4.3.2  Localisation

We implement domain localisation as described in Bocquet (2016, his Tab. 2). In the domain localisation, each model grid
point (local domain) assimilates observations within a circle centred on the grid point. To gradually reduce the impact of
observations away from the central grid point and to insure spatially smooth DA corrections, the observation error is tapered
by the Gaspari-Cohn (GC) function with a cutoff value of $10^{-5}$. In this way, each local domain has a different cost function
based on different observations. Moreover, the localisation radii are dependent on the observation type. The choices that we
implement for the localisation radii in this study are given in Tab. 3 while their rationale and impacts are discussed in Sect. 5.1.

We anticipate that the localisation radius for global parameters (e.g. drag coefficient, damage parameter, etc.) is set to infinity.
This choice reflects the fact that the parameter does not have a spatial de-correlation scale and is indeed global. This treatment
of localisation for global parameters is also adopted in Aksoy et al. (2006); Ruiz et al. (2013b); Massonnet et al. (2014). See
also Ruckstuhl and Janjić (2018); Bocquet et al. (2021); Malartic et al. (2022) for more recent developments on this topic.

### 4.4  Treatment of bounded physical variables and model parameters

Of the nine MEB model variables, three are bounded: SIC, the level of damage and SIT. Inferring them via the IEnKF is
therefore challenging given the Gaussian assumption on which the IEnKF based. In practice, there is no guarantee that the
update analyses of SIC, damage level or SIT, fall within their bounds.

Hence, the solution to that problem that we adopt in this study, albeit sub-optimal, is very pragmatic and straightforward.
When the analysis of these variables falls outside of their bounds, they are forcibly set to their nearest bounds. We are conscious
that this approach leads to local ensemble collapse (whereby some members originally having out-of-range analysis values
area made all equal to the nearest boundary value) and can cause biases in the analysis. Nevertheless, the results in Sect. 5





will demonstrate that, when the bounds are not exceeded too often, the approach works well, it does not cause major ensemble collapse and it is successful in removing non physical values.

We follow a similar strategy when performing the estimation of the damage parameter, $\alpha$. There the analysed value of $\alpha$ assumed bounded from below by $\alpha = 2$, the value at which the analysed $\alpha$ will be taken if lower than 2.

Our pragmatic approach is however insufficient when estimating the drag coefficient $C_a$. This parameter is bounded to be

strictly positive, therefore ensemble collapse could happen whenever the ensemble mean of $C_a$ approaches zero. In that case a potentially large number of members may get negative analysis values that would all be restored to the same little positive values. Hence, we adopt a re-sampling approach in which, with each negative analysis of $C_a$, we sample from $\mathcal{N}(10^{-5}, 10^{-6})$ until a positive value is obtained.

## 5   Results

We evaluate the performance of the IEnKF for state and parameter estimation in the dynamics-only MEB sea ice model under the four different scenarios described in Tab. 3. We use the root-mean square error (RMSE) over both time- and space as skill metric.

### 5.1   Optimising the localisation radius

As discussed in Sect. 4.3.2, the domain localisation allows for different localisation radius per observation type. The localisation

radius is related to the spatial correlation scale of the model physics. The latter is naturally time- and space-dependent. In practice, for the sake of computational efficiency and for the complexity of its adequate assessment, the localisation radius is often a fixed value (albeit dependence on the physical variable is usually accommodated).

As a trade-off between efficiency and accuracy, we opted for optimising the localisation radius when only one model field is observed. Furthermore, we assume that the SIV has the same correlation length scale along both components (i.e. no preferred

direction of movement), consistently with the experimental setup.

For each of the localisation radius explored we run a $15\,\mathrm{days}$-long simulation, after a $42\,\mathrm{days}$ ensemble free run without DA. Although $15\,\mathrm{days}$ may appear as too few for an adequate tuning and may not include all possible physical regimes, it still covers multiple storm events (storms passage every $3.5\,\mathrm{days}$ on average).

The ratio between the RMSE of the analysis and of the free run, as function of the localisation radius when assimilating

either SIT, SIC or SIV, is shown in Fig. 3. When assimilating SIT (Fig. 3a), the analysis error in SIT decreases monotonically with the localisation radius until the localisation radius reaches the domain size ($200\,\mathrm{km}$). The inset shows that assimilating SIT leads also to a reduction in SIV's RMSE ($\mathrm{RMSE}^{\mathrm{a}}/\mathrm{RMSE}^{\mathrm{free}}$ systematically below one), while it instead leads to a deterioration in the SIC as soon as the localisation radius is larger than $5\,\mathrm{km}$.

As opposed to when SIT is observed, the localisation radius for SIC observations (Fig. 3b) is very small. After $5\,\mathrm{km}$, the

analysis error in SIC increases monotonically with the localisation radius. Observing SIC improves both SIT and, for very long radii, also SIV (see inset in Fig. 3b).





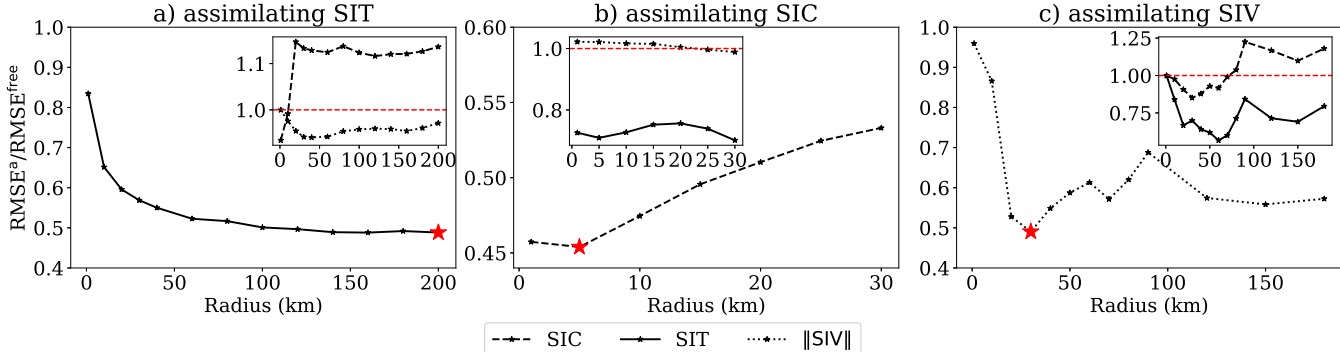

**Figure 3.** $\text{RMSE}^{\text{a}}/\text{RMSE}^{\text{free}}$ as a function of the localisation radius, in experiments with observed SIT (a), SIC (b) or SIV (c). The red star indicates the lowest analysis error. The insets show the error of other two unobserved model fields out of the three observed model variables, and the red dashed line indicates the point where $\text{RMSE}^{\text{a}} = \text{RMSE}^{\text{free}}$.

Finally, from Fig. 3c we see that the lowest analysis RMSE when assimilating SIV is attained with a localisation radius around $30 \, \text{km}$ and that both SIC and SIT will improve in the multivariate update.

Based on these results, the most effective localisation radius for the observed model fields are: $200 \, \text{km}$ for the SIT observations, $5 \, \text{km}$ for the SIC observations and $30 \, \text{km}$ for SIV observations. These different localisation radii arise from the physical spatial correlations and the observation density of these model fields.

## 5.2  Scenario 1: Inferring the model physical variables under a perfect model setup

Here we study in detail the fully multivariate DA using different combinations of observations under a perfect model scenario. The RMSE is calculated over $30 \, \text{days}$-long assimilation experiments that follow a $42 \, \text{days}$ free ensemble run without DA. In the $30 \, \text{days}$ assimilation, we assimilate observations daily as mentioned in Sect. 4.2 leading to a total of 31 analyses.

As mentioned in Sect. 4.4, three out of the nine model fields are bounded quantities: SIC, SIT and the level of damage. Given that the respect of those bounds is not automatically guaranteed by the DA procedure, we apply a post-processing step. Here, we quantify how often nonphysical values (i.e. values out of the bounds) are produced in the analysis: Tab. 6 shows the number of violations during the 30-day DA period.

When only SIC is assimilated, physical bounds for SIC and for the level of damage are exceeded very occasionally, below $1.5\%$ of the total analyses and grid points. On the other hand, when SIC is not observed, the chance of getting nonphysical SIC analyses increases by two order of magnitudes, although it remains below $5.2\%$. Similarly, the SIC observations lower the chance of getting nonphysical damage analyses. Notably, when SIT is observed it leads to analyses that more often violate the physical bounds, particularly the upper bound for the level of damage. The damage bound violations are more severe than those in SIC since most of the time the sea ice is undamaged and thus very close to the potentially violable bounds of the model fields. Without the possibility of observing the damage field, the cross-correlations may amplify the analysis increments. This

**Table 6.** The total percentage of local analyses that violate the physical bounds in the 30-day multivariate DA experiments.

| Assimilated observations | Physical bounds violation | | | |
|:---:|:---:|:---:|:---:|:---:|
| | $SIC < 0$ | $SIC > 1$ | $d < 0$ | $d > 1$ |
| SIC | 0.01% | 0.01% | 0.02% | 1.41% |
| SIT | 0.25% | 6.95% | 0.34% | 18.66% |
| SIV | 0.23% | 1.36% | 0.19% | 11.80% |
| SIC+SIT | 0.01% | 3.45% | 0.09% | 20.93% |
| SIC+SIV | 0.01% | 0.85% | 0.17% | 12.09% |
| SIT+SIV | 0.22% | 5.17% | 0.24% | 18.87% |
| SIC+SIT+SIV | 0.01% | 3.75% | 0.17% | 20.64% |
| SIC+SIT30+SIV | 0.01% | 0.98% | 0.14% | 12.97% |

*The bound $SIT > 0$ is never violated because SIT is always larger than 1 m in our experiments.

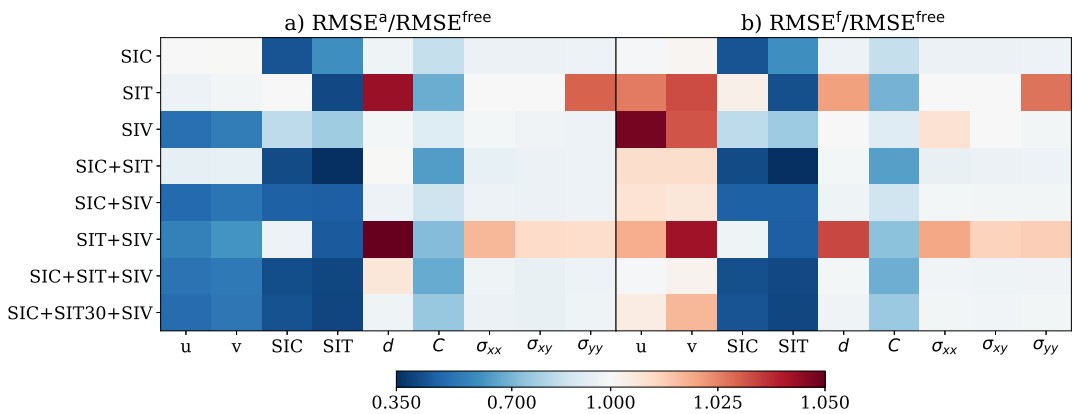

**Figure 4.** RMSE in the state estimate for *scenario 1*. (a) RMSE$^a$/RMSE$^{free}$ (b) RMSE$^f$/RMSE$^{free}$. Columns display the individual physical model fields; rows refer the (combination of) observations that are assimilated.

effect can be mitigated by limiting the localisation radius of SIT to 30 km as shown in the row of "SIC+SIT30+SIV". The violation of the physical bounds is efficiently addressed by the post-processing step that brings them within their physical limits as described in Sect. 4.4 and it does not lead to an increased RMSE afterwards. As shown in Fig. 4, the SIC analysis is 380 improved compared to the free run in all cases and the sea ice damage is the only model field that can be less accurate than the free run.

Figure 4 shows the ratios analysis/free-run (a) and forecast/free-run (b) RMSE (averaged in space and time) for different type of observations (column-wise) and variable (row-wise). Values smaller than 1.0 indicate that DA brings in general an improvement in the state estimate compared to the free run. Comparing panels (a) and (b) we can evaluate how much of 385 the DA update is effective in reducing the forecast error (recall that the analysis cycle is 1 day long). From the figure we





**Table 7.** Time-averaged spectral norm (the largest singular value of a matrix) of the cross-correlation matrices between selected model fields when all observations are assimilated. Here the $u$-component of SIV is chosen to be shown.

| Corr($u$, $\sigma_{xx}$) | Corr($u$, $d$) | Corr($u$, SIC) | Corr($u$, SIT) | Corr(SIC, SIT) |
|---|---|---|---|---|
| 76.99 | 67.35 | 41.00 | 42.03 | 59.72 |

immediately notice that when only one observation field is assimilated, that same field gets most of the improvement in the analysis. This is consistent with our results in Sect. 5.1. Figure 4 also shows improvements in SIV (SIC) when SIC (SIT) alone is assimilated. This appears to contrast with Fig. 3, which displayed a slight deterioration in the same case (cf. Fig. 3b and c insets). Nevertheless, the longer experiments of Fig. 4 suggest that the ensemble has acquired dynamical consistency and therefore better reproduces the cross-variables correlations.

In the MEB rheology, the level of damage, cohesion and stress have a close relationship. Without assimilating SIT, the IEnKF-N leads to improvements in sea ice cohesion and stress and some improvements in the level of damage too. However, the assimilation of SIT tends to result in overly damaged sea ice. This is also evident from Tab. 6 where the assimilation of SIT leads to higher chances of breaking the bounds of the level of damage. This adverse effect on the damage and stress persists when SIT is assimilated together with SIV. Interestingly, however, when SIT is assimilated together with SIC, the adverse effect is subdued. An ad-hoc remedy is to use a shorter localisation radius for SIT (experiment "SIC+SIT30+SIV" in Fig. 4), although it also reduces the improvements in SIT and cohesion. Moreover, as a result of better cross-correlations, the use of a small localisation mitigates the violations of physical bounds as shown in Tab. 6.

Due to the importance of cross-correlation derived from the forecast ensemble, we show the strength of the cross-correlation matrix between different model variables measured by a spectral norm as shown in Table 7. Here the spectral norm is defined as the largest singular value of a cross-correlation matrix. Note that the spectral norm is an overall measure of the magnitude of a matrix and is always positive. We use it here as a means of comparison of the strength of the cross-correlation between two variables. These cross-correlations can be understood from a physical point of view. The cross-correlations between the SIV and stress are the strongest as SIV is mainly driven by the external wind field. In addition, SIV, stress and level of damage are closely coupled processes, so the level of damage and the SIV also shows strong cross-correlation. The weak cross-correlation between SIV and SIC/SIT is a result of the small magnitude of SIV which transports SIC, SIT and cohesion. Nevertheless, this gives rise to a strong correlation between SIC and SIT because they are both controlled by the advection processes.

A physical interpretation can also be invoked to explain why the improvements in SIV analysis does not necessarily translate into improvements in the forecast as shown in Fig. 4. We argue that this is due to the instant injection of error from the wind field after the assimilation of observations. The correction from the assimilation acts as a perturbation to the model which induces a model adjustment. In contrast to SIV, the increased error in the analysis of the level of damage, when assimilating SIT, is mitigated by the further damage caused by the wind field. On the other hand, the longer timescale in the variability of SIC, SIT and cohesion makes them less sensitive to the model adjustments and thus better analysis generally yields better sea ice forecasts (cf. Fig. 4).



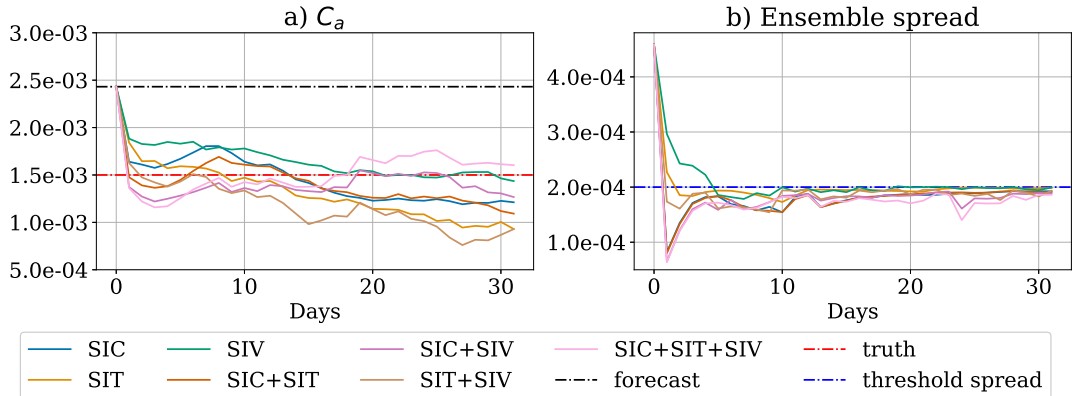

**Figure 5.** Time series of a) the analysis of $C_a$ using different combinations of observations; b) the analysis ensemble spread of $C_a$;

In summary, our experiments show that the IEnKF can improve both the analysis and forecast of the MEB sea-ice model. We confirm, in line with previous studies, the necessity of controlling the main source of the uncertainty from the external forcing in the sea ice model. We argue that a correct wind field can counteract the deterioration of the inaccurate SIV forecast. Furthermore, the results also demonstrate the positive impact of a variable-dependent localisation radius to combat both sampling error and nonlinearities. Moreover, even if the IEnKF suffers from sampling error with a large localisation radius, the deterioration of some of the unobserved fields is marginal with the RMSE being only $6\%$ larger than the free run. With a reduced localisation radius, the IEnKF improves all the unobserved fields.

### 5.3 Scenario 2: Inferring the model physical variables and the drag coefficient $C_a$

We achieved satisfactory performance in the state estimation of the MEB model using the IEnKF-N under the perfect model assumption. In the experiments described in this section, we assume that $C_a$ is incorrectly specified and attempt to recover the true value using DA, while all other parameters are perfectly known.

The air drag coefficient, $C_a$, controls the degree to which momentum from the wind is transferred to the sea ice cover. In our model, similar to most sea ice models, it is a constant scalar value that modulates the wind drag, cf. Eq. (2). Equation (2) shows two main sources of uncertainties in the same term: the wind field, $\mathbf{u_a}$, and the drag coefficient $C_a$. The multiplicative role of $C_a$ makes it an amplification factor on the uncertainty originated from uncertain wind fields. Given that the wind field is the main source of uncertainties in the sea ice dynamics, the incorrect specification of $C_a$ affects also the predictability of the sea ice. To see this, note in Tab. 5, the larger free run ensemble spread in this scenario compared to the perfect model scenario. Note that the difference in the $C_a$ between scenarios 1 and 2 is as small as $10^{-3}$ (cf. Tab. 3).

Similar to the state estimation experiments, the assimilation starts after a $42$-day free ensemble run. Recall that here we do not use adaptive inflation and a fixed multiplicative inflation factor of $1.02$ for physical variables are used (cf. Tab. 3). For the parameter $C_a$, we adopt a time-dependent inflation based on Eq. (7).





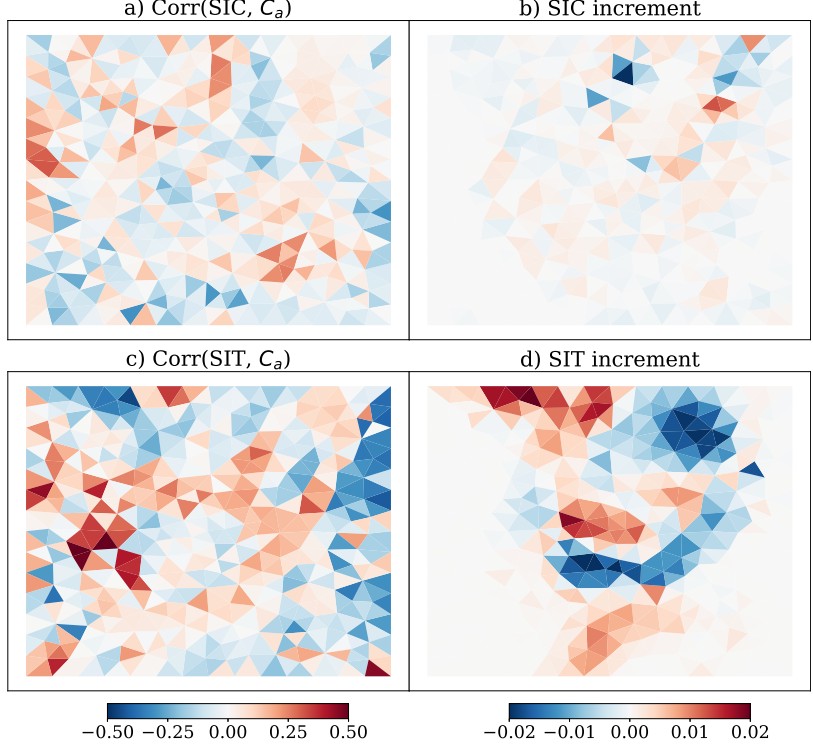

**Figure 6.** Cross-correlation between the observation and $C_a$: a) correlation between SIC and $C_a$ at the last analysis; b) SIC increment. In both cases it refers to experiment when only SIC is assimilated; c) and d) the same as (a) and (b) but for SIT in place of SIC in experiment when only SIT is assimilated.

Figure 5a shows the time series of the analysis of $C_a$, while Fig. 5b shows the analysis spread in $C_a$ as a function of time. We see that after the drastic correction of the initial bias, the ensemble spread stabilises to the threshold of the ensemble spread, $\sigma_{\mathrm{std}}^t$. A first remarkable feature is that in all experiments, $C_a$ drops significantly over the first time steps, thereby approaching (but not necessarily converging to) the true value (red line). This is a clear consequence of a strong correlation between the observed fields and the parameter. Besides this, we then observe different converging value and performance depending on the type of observations assimilated. When only one type of observation is assimilated, either SIC or SIT, the analyses underestimate $C_a$ at the end of our experiment time. In particular, although assimilating either SIC or SIT alone leads to an underestimated $C_a$, assimilating SIC gives $C_a$ values closer to the truth with progressively smaller increments, a signature of convergence. The smaller increments of $C_a$, when assimilating SIC, is a result of smaller cross-correlation between SIC and $C_a$ in comparison to SIT and $C_a$ as shown in Fig. 6a and c. With the same ensemble spread of $C_a$, the cross-correlation reflects a large ensemble spread of SIT. Similar to the increment of $C_a$, the SIT experiment also shows increased increments of the observed fields compared to the SIC observation as in Fig. 6b and d. Moreover, the cross-correlation of the ensemble anomaly between SIC/SIT and $C_a$ is spatially inhomogeneous (cf. Fig. 5a and c). This implies that the error is not controlled solely



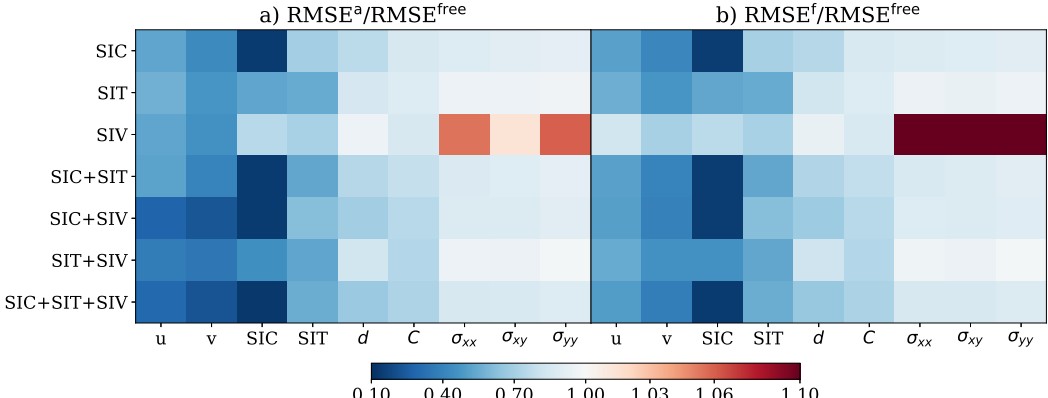

**Figure 7.** RMSE in the state estimate for *scenario 2*. (a) RMSE$^a$/RMSE$^{free}$ (b) RMSE$^f$/RMSE$^{free}$. Columns display the individual physical model fields; rows refer the (combination of) observations that are assimilated.

by the global parameter but also by other spatially dependent processes. One possibility is the error in the wind fields. As the ensemble error is primarily driven by the error in the wind field scaled by $C_a$ acting as wind forcing, the cross-correlation between the $C_a$ and the observations may be affected by the error in the wind fields. This suggests that while the IEnKF successfully corrects large biases in $C_a$ it may not be able to correct equally well errors of smaller magnitude as the errors can be aliased with the error in the wind field. In the latter case, the estimation of $C_a$ does not necessarily converge to the "true" $C_a$.

The correction of $C_a$ is also effective in the state estimation as shown in Fig. 7. The IEnKF efficiently controls the RMSE for nearly all model fields regardless of the combination of observations compared to the free run that is instead affected by a large bias in $C_a$. Notably, compared to the free run, SIV is improved not only in the analysis, but also in the forecast. This is different from the multivariate update in Sect. 5.2 because the positively biased $C_a$ amplifies the uncertainties from the wind field which is then reduced by the corrected $C_a$. The time series in Fig. 8c-d show that the corrected $C_a$ significantly reduces the bias in SIV, but the transient error in the wind forcing still impacts the accuracy of the SIV forecast.

In Fig. 5a, the best skill in estimating $C_a$ is achieved when assimilating SIV alone due to its close relationship with the wind field and $C_a$ in Eq. (2). Notably, assimilating only SIV greatly reduces the RMSE across all model fields except for the sea ice stress (cf. the third row in Fig. 7a). The increased RMSE in the sea ice stress arises because the DA update of the stress sometimes severely violates the constitutive equation. This can potentially lead to unstable model solutions and model crashes. Model crashes can be avoided by using a large number of iterations for the MEB solver during an extended periods after the DA step, but the unphysical update still leads to inaccurate forecasts of the sea ice stress (cf. the third row in Fig. 7a). The inaccurate stress forecast can be observed in Fig. 8a, which shows a spike in the stress time series. Such a spike does not occur when all observations are assimilated as shown in Fig. 8b. We found that the erroneous stress spike in the time series occurs when the local element of SIC is reduced by the assimilation (not shown) to the point that it creates large SIC gradient and increases the elastic behaviour of the sea ice (see the right hand side of Eq. 3). Since the temporal variation of the SIC is small,





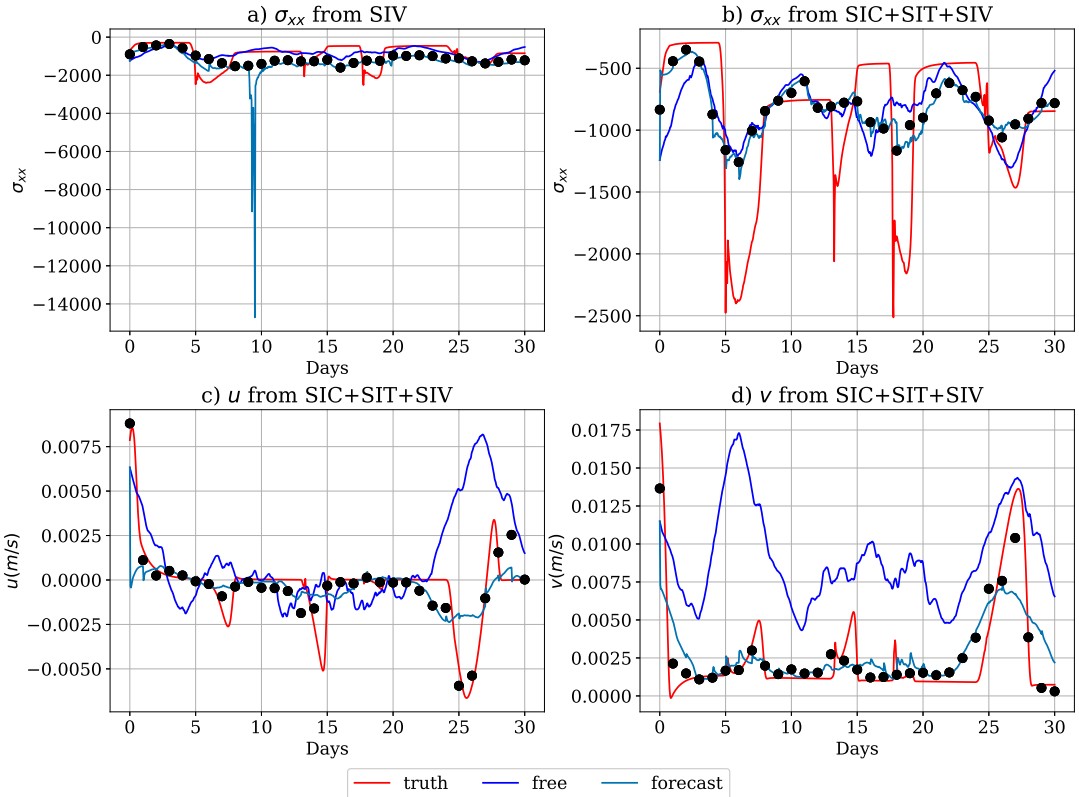

**Figure 8.** Time series of a) sea ice stress in the x direction when only SIV is assimilated; b) sea ice stress in the x direction when all observed fields are assimilated; c) SIV in $u-$ and d) $v-$component when all observed fields are assimilated. The black points in the time series are the spatially averaged analysis.

this issue persists, and maintains a continued decrease in the sea ice stress along with the model integration. This incorrect SIC estimate is remedied in the next DA step where the multivariate DA restores the SIC. Hence, assimilating SIC mitigates the unphysical DA update. Arguably, it is unlikely that only the SIV is assimilated in real scenarios, yet our results suggest that it is wise to restrict the multivariate update of the sea ice concentration in this case.

475    Our experiments show that the IEnKF is able to reduce the bias in $C_a$ based on available sea ice observations. The improved $C_a$ estimation significantly reduces the error in the model fields. From the momentum Eq. (2), we know that the SIV is directly linked to $C_a$. Our results show the importance of assimilating SIV for estimating $C_a$. SIC and SIT, though less effective than SIV, still improves the estimate of $C_a$ and of the state. Importantly, when assimilated in conjunction with SIV, SIC and/or SIT mitigate the imbalance of the constitutive equations.



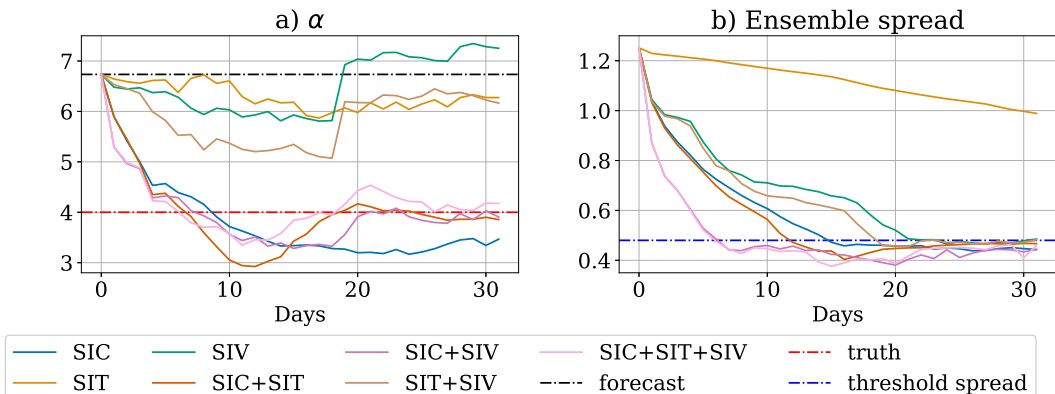

**Figure 9.** Time series of a) the analysis of $\alpha$ and b) its ensemble spread.

### 5.4 Scenario 3: Inferring the model physical variables and its erroneous damage parameter $\alpha$

While the drag coefficient, $C_a$, is linked to the external forcing, the internal property of the sea ice is largely controlled by the damage parameter $\alpha$. The damage enters the model in the stress Eq. (3). Although $\alpha$ is added to the model in an ad-hoc manner, it plays an essential role in the MEB rheology as it sets the rate at which viscosity decreases with increasing level of damage and thereby controls the transition between the elastic-brittle regime at low damage and the viscous regime at high damage.

Model sensitivity studies by Dansereau (2016); Weiss and Dansereau (2017) showed that the value of $\alpha$ is critical in determining the macroscopic mechanical behaviour of the model, and that a value of $\alpha \geq 4$ leads to complex, sea-ice compatible, behaviours. In fact, in our experiments, the truth, i.e. the un-biased model, is set to $\alpha = 4$ (cf. Tab. 3). In the DA experiments with estimate of $\alpha$, we mimic an initial biased estimation of the parameter that is in the same range of sea-ice compatible behaviour: we choose $\alpha = 6.5$ (cf. Tab. 3). Our strategy is realistic as the initial "guess" does not cause the model to behave qualitatively differently from the observations when $\alpha \geq 4$. Drastic changes of the dynamical regimes are also challenging for DA and it influences the error dynamics between the model parameter and the observed fields. Given that the IEnKF updates (both in the state fields and parameters) are unbounded, we apply the same post-process to keep the analyses within physically acceptable bounds as described in Sect. 4.4. Finally, similar to the experiments with biased $C_a$, we apply a fixed inflation factor of $1.02$ to the model state, and a time-dependent inflation to the parameter uncertainty based on Eq. 7 (cf. Tab. 3).

One of the fundamental challenges in estimating $\alpha$ arises from the nonlinear relationship between $\alpha$ and the observed fields. In particular, the parameter is directly related to the stress field, which is not observable. The complex, nonlinear and indirect nature of these relations, can lead to inaccurate (finite) ensemble-based cross-correlations. Another potential challenge comes from the little sensitivity of the model fields to $\alpha$. As shown in Tab. 5, in a 90-day free run, the ensemble spread of observed model fields is only marginally larger than that in the perfect model scenario.

Despite these obstacles, the IEnKF shows encouraging results in estimating $\alpha$ as shown in Fig. 9a. Assimilating SIC or SIT alone leads to under- and over-estimation of $\alpha$ after 30 days. From Fig. 9b, we see that the ensemble spread in SIT is



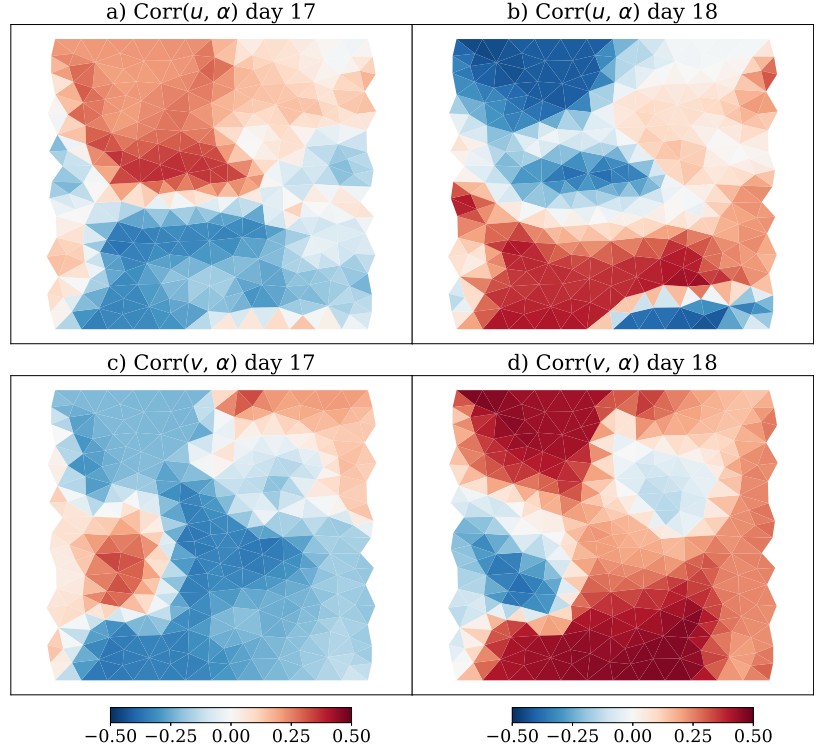

**Figure 10.** Cross-correlation between (a,c) $u$- and (b,d) $v$-component of SIV and $\alpha$ at day 17 and 18 when only SIV is assimilated.

still relatively large at the end of the experiment. This suggests that it is only slightly reduced at the analysis steps but also that further adjustment of $\alpha$ beyond the 30-th day is still a possibility. The simultaneous assimilation of SIT and SIC leads to an almost full recovery of the true value of $\alpha = 4$. Similar results are attained whenever SIC is assimilated (cf. experiment

SIC+SIV or SIC+SIT+SIV). This suggests the crucial relationship between $\alpha$ and SIC, which appears as the key observation for inferring the damage parameter.

Our results suggest also that observations of SIV cannot be used to retrieve $\alpha$ effectively. In all the experiments with SIV observations, the estimated $\alpha$ gradually approaches the truth until $\mathrm{day}$ 18, when it then abruptly diverges away from it. An insight on the reasons behind this sudden change is provided in Fig. 10, which shows the spatial distribution (on the model

domain) of the cross-correlation between $\alpha$ and either $u$ or $v$, at days 17 and 18. The cross-correlation between SIV and $\alpha$ flip their signs spatially from day 17 to 18. This is related to the uncertainties in the wind field which dominates the forecast uncertainty in the SIV.

Let us illustrate this issue with simple mathematical arguments. We assume that the nonlinear sea ice model, $\mathcal{M}$, can be approximated by its linearisation $\mathbf{M}$ within the forecast interval in between two successive analyses. The forecast ensemble

anomaly (error) of SIV at time step $k$ can be approximated as

$$\delta\mathbf{u}_k = \mathbf{M}_k^\alpha \delta\alpha_{k-1} + \mathbf{M}_k^{u_a}\delta\mathbf{u}_{\mathbf{a}\,k-1} + \mathbf{M}_k^x \delta\mathbf{x}_{k-1} + \mathcal{O}(2) \tag{8}$$



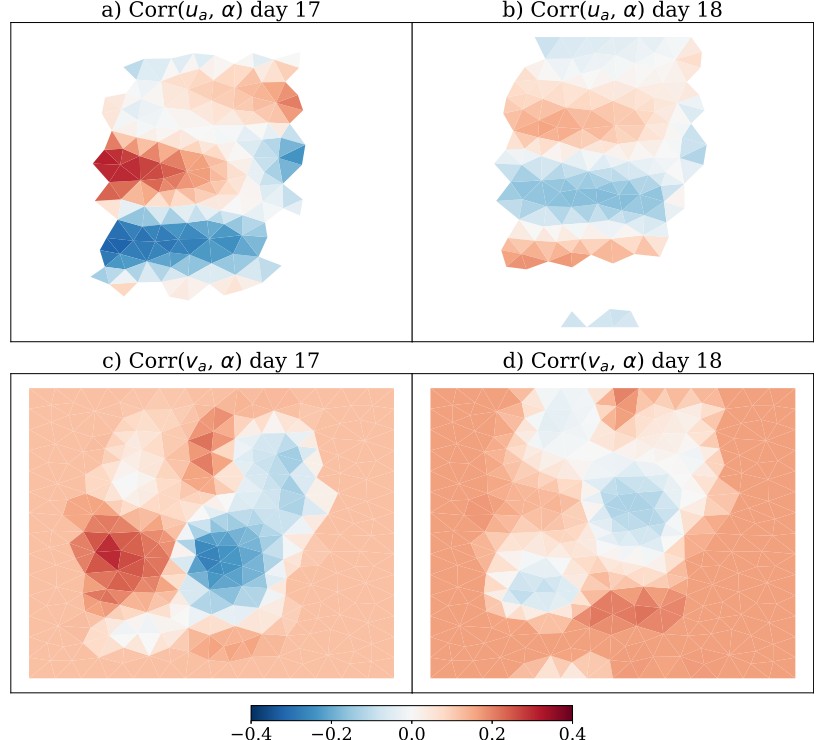

**Figure 11.** Same as Fig. 10 but for the components of the wind forcing in place of those of SIV.

where $\delta\cdot$ represents the deviation from the ensemble mean, and $\mathbf{x}$ is the state vector except the damage parameter, the superscript means the model sensitivity to the corresponding variables in which $\mathbf{M}_k^\alpha \in \mathbb{R}^{n\times 1}$, $\mathbf{M}_k^{u_a} \in \mathbb{R}^{n\times m}$, $\mathbf{M}_k^x \in \mathbb{R}^{n\times l}$ with $n$ being the number of SIV in the controlled vector, $m$ being the number of elements in the wind fields vector and $l$ being the number of

elements of model state in the controlled vector. The $\mathcal{O}(2)$ represents the high order terms that are greater than 2nd order. In the EnKF (and thus in the IEnKF), the cross-covariance is estimated from the ensemble anomaly of the SIV and the perturbations of $\alpha$, which approximates $\mathbb{E}[\delta\mathbf{u}_k \delta\alpha_k]$. This cross-covariance between SIV and $\alpha$ is related to the perturbation (error) from the model state, wind and $\alpha$. In our case, whenever $||\mathbf{M}_k^{u_a}\delta\mathbf{u}_a \gg \mathbf{M}_k^\alpha \delta\alpha||$, the SIV uncertainty is mainly driven by the wind. This can falsely give a strong cross-correlation between SIV and $\alpha$ producing incorrect $\alpha$ estimate when it is only based on

SIV observations. A similar issue was encountered in Simon and Bertino (2012) where they found the parameter estimation challenging when the uncertainty of the parameters show relatively little uncertainties on the observed fields.

     Based on this argument, we can show that the sign flip of cross-correlations in Fig. 10 is a result of the change in wind field. In Fig. 11, the cross-correlation between the wind field and $\alpha$ shows the periodic northward travel of the storm in $y$-axis (cf. Fig. 11a and b) and the rotation of the storm (cf. Fig. 11c and d). This matches with the changes of cross-correlations in

Fig. 10. The incorrect estimate of $\alpha$ highlights the challenges when the primary sources of the uncertainties are external instead



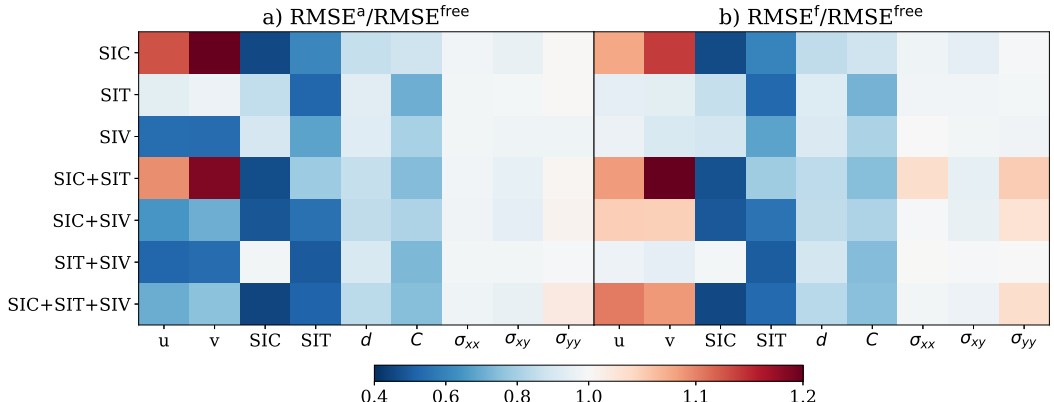

**Figure 12.** RMSE in the state estimate for *scenario 3*. (a) RMSE$^a$/RMSE$^{free}$ (b) RMSE$^f$/RMSE$^{free}$. Columns display the individual physical model fields; rows refer the (combination of) observations that are assimilated.

of being the model parameters. We observe that this effect also influences the $\alpha$ estimate when the SIV is assimilated with the SIT.

Figure 12 shows the improvements in the state estimate. As expected, the analysis of the observed fields are in general more accurate than the free run. Moreover, compared to the perfect model scenario, the damage field is now improved relative to

the free run in all experiments. The stress field is only moderately improved even though it has direct relationship with $\alpha$ in Eq. (3).

Interestingly, without observing SIV, the assimilation of SIC shows a negative impact on the analysis of SIV. This may be a result of the underestimated $\alpha$ in these experiments. With low value of the $\alpha$, the sea ice has a more elastic behaviour. The elastic motion has a short timescale and is sensitive to the perturbation of the sea ice variables making the slowly changed

SIC observations unreliable. This leads to deteriorated SIV forecast even if SIV is assimilated with SIC. On the contrary, assimilating either SIT or SIV leads to improvements in SIV forecast. This is consistent with the finding from Weiss and Dansereau (2017) such that, with high $\alpha$ values, with the sea ice transitioned from elastic to viscous behaviour the system becomes more predictable.

Our results demonstrate the possibility of estimating $\alpha$ successfully using as many as possible of the available observations.

It is notable that not a single type of observation alone can infer $\alpha$ accurately. We also demonstrate that the IEnKF cannot always identify the correct source of error in a complex environment. In our case, the error from the wind field negatively impacts the estimation of $\alpha$. Interestingly, a deteriorated $\alpha$ analysis does not necessarily lead to a deteriorated state estimation. In contrast, the overestimation of $\alpha$ still moderately improves the sea ice forecast due to improved predictability.



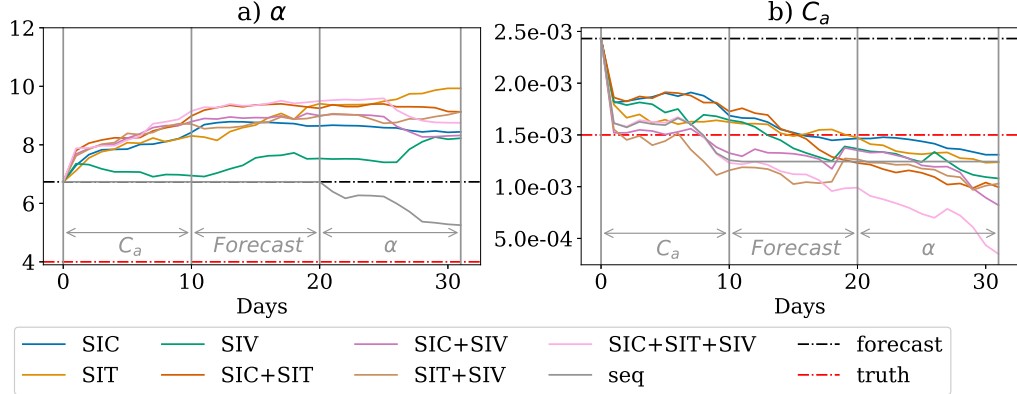

**Figure 13.** The analysis of $\alpha$ (a) and $C_a$ (b) parameter estimate based on a variety of combinations of observations.

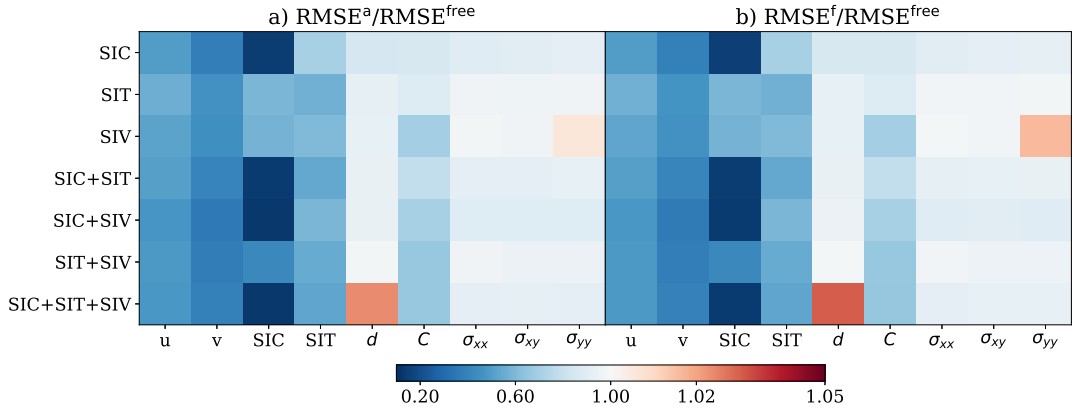

**Figure 14.** RMSE in the state estimate for *scenario 4*. (a) RMSE$^a$/RMSE$^{free}$ (b) RMSE$^f$/RMSE$^{free}$. Columns display the individual physical model fields; rows refer the (combination of) observations that are assimilated.

## 5.5 Scenario 4: Inferring the model physical variables and its erroneous $C_a$ and $\alpha$

In the previous sections we demonstrated that, with sufficient observations, the IEnKF can estimate $C_a$ and $\alpha$ when only one of them is erroneous. Considering that both $C_a$ and $\alpha$ are in the closely related equations for SIV and stress (cf. Eq. (2) and (3)), it is of interest to investigate the possibility of estimating both model parameters simultaneously.

    Figure 13b shows the estimated $C_a$ after $30$ days of assimilation. All experiments underestimate $C_a$ (although starting from an overestimated value) and shows no sign of convergence with time. In particular, when all types of observations are

assimilated, the estimated $C_a$ is the farthest from the truth. The underestimation of $C_a$ is accompanied by an increase of the estimate of $\alpha$ as shown in Fig. 13a. Although this demonstrates the difficulty in estimating both parameters, it is remarkable that it still leads to an improved forecast of model fields as shown in Fig. 14. This is known as a compensating effect (Bocquet, 2012). This apparent contradiction is related to the non-identifiability of the problem whereby the parameter estimation cannot




fully reconstruct the true parameters. In practice, this implies that more than one set of parameters can produce the same observed fields. The identifiability issue can also be viewed from a physical standpoint: $C_a$ determines the motion of the sea ice via the wind forcing. With reduced $C_a$, the wind can cause a slower sea ice motion. $\alpha$ influences the rate at which sea ice transitions from an elastic-brittle solid behaviour to a viscous fluid behaviour with increasing level of damage. As discussed in Sect. 5.4, the increased $\alpha$ leads to more viscous sea ice that is more easily subjected to permanent deformations. With high $\alpha$ and low $C_a$, the sea ice is deformed by the wind forcing with weakened oscillatory elastic behaviour. This means that the oscillatory elastic motion of the sea ice is replaced by the motion driven by the wind forcing which can lead to similar sea ice properties as the truth.

The identifiability issue can also be associated with incorrect cross-correlations between model fields and model parameters from different sources of errors as described in Eq. (8). We observe that, assimilation correctly decreases $C_a$ showing a relatively correct cross-correlation between the observations and $C_a$. This controls the impact of the external wind forcing as a positively biased $C_a$ can amplify the uncertainty in the wind field, which increases the term $\mathbf{M}^{u_a}\delta\mathbf{u}_a$ in Eq. (8). As discussed in Sect. 5.4, the outstanding external wind forcing could lead to erroneous estimation of $\alpha$. Hence, after the external uncertainties are controlled, the ensemble should be able to develop reasonable cross-correlations between $\alpha$ and the observations. To test this, we adopt the following strategy in the 30-day assimilation experiments: 1) Constraining the external uncertainty by estimating $C_a$ only for 10 days; 2) developing uncertainty from $\alpha$ by model forecast without assimilation for 10 days; 3) estimating $\alpha$ for the last 10 days. This is labeled as "seq" in Fig. 13. Although $C_a$ is still underestimated, we observe a reduction of error in $\alpha$. Also, as other experiments do not show a decreased $\alpha$ after an underestimated $C_a$, this may suggest the importance of step 2) where the uncertainty arising from $\alpha$ developed. We stress here that the 10 days is simply chosen for the convenience and is not tuned.

Our results show that the augmented state vector can have identifiability issues when both $C_a$ and $\alpha$ are biased. As shown in Tab. 5, the perturbation of $\alpha$ cannot provide greater forecast uncertainty compared to the positively biased $C_a$. We have proposed a strategy to overcome the issue with some discussions on alternative approaches. For example, a larger ensemble spread of $\alpha$ may increase the uncertainty in the forecast ensemble which avoids a single dominant source of forecast uncertainty. We note that, based on our reasoning, the identifiability issue may not exist when $C_a$ is negatively biased. Nevertheless, this demonstrates a potential issue in the parameter estimation in MEB-type sea ice model.

## 6 Discussion

In this study, we investigate joint state and parameter estimation of a MEB sea ice model using IEnKF. This study focuses on the cross-covariance between model fields which is crucial for the correct parameter estimation. Given that the IEnKF is also a variational method, it minimises the cost function in Eq. (1) and thus it adjusts the model parameter at each Gauss-Newton iteration. The adjusted, data-informed, model parameters allows for a natural treatment of the nonlinearity thanks to the execution of the model within one analysis step or, had we used the IEnKS, within the entire assimilation window. The





variational formulation has also other additional advantages over the traditional EnKF. For example, one can impose constraint optimisation and regularisation to the cost function to avoid numerical problems or to append physical constraints.

One important ingredient of IEnKF is the use of ensemble. The forecast ensemble of the IEnKF can suffer from ensemble collapse as discussed in Sect. 4.3 where inflation is used as an effective modification to the forecast ensemble. In this study,

two different inflation approaches are used. In the perfect model scenario, we adopt an adaptive inflation method, the IEnKF-N (Bocquet and Sakov, 2012). The IEnKF-N works very well when only the system's state (its physical variables) are inferred. On the other hands, its performance in the state and parameter estimation case (not shown here) is unsatisfactory. We argue that this is related to the nature of the other key fix for EnKF, the localisation. In our experiments, the localisation in the IEnKF is done in the domain. In practice, each grid point is updated individually, using the observations gathered within a given local

domain. The adaptive inflation of IEnKF-N is therefore itself dependent on the model field and grid point. To see this, let us write the analysis error covariance after the adaptive inflation as $\mathbf{P}^{\mathrm{a}} \to \mathbf{W} \circ \mathbf{P}^{\mathrm{a}}$ with each entry of the matrix $\mathbf{W}$ containing the field-specific inflation factor, and $\circ$ being the Schur product. As a result, the cross-correlation across different model fields and spatial points described in the un-inflated $\mathbf{P}^{\mathrm{a}}$ are modified by the inflation, potentially breaking the physically and dynamically sounds correlation developed in the ensemble.

This undesired effect of the inflation does not cause trouble when only the model state is estimated because the domain localisation acts exclusively on the cross-correlations within the localisation radius, thus limiting the impact of the weighting from the inflation factor. In the (global) parameter estimation problem on the other hand, localisation is not used and thus the effect of the inflation factor in breaking the cross-correlation between the observations and the parameters becomes more relevant. We foresee that this aspect of the IEnKF-N may limit is direct applicability when dealing with global parameter estimation in

high-dimensional systems, whereby using localisation is mandatory. Further investigations is being undergone to address this issue, including the development of an alternative inflation and/or localisation strategy, e.g. one proposed by Pasmans et al. (2023).

In this study we have used a spatially homogeneous and constant inflation factor for the model state, but for the parameters we have adaptively inflated the error ensemble-based covariance to maintain the forecast ensemble larger than a given bound

(cf Sect. 5.3). Although we have not performed any fine tuning of this inflation factor (i.e., for the lower bound for the ensemble variance) our results are promising and left room for further improvement.

We also shed lights on potential issues when the forecast uncertainty is driven mainly by the external wind field. In this case, the cross-covariance matrix reflects the error from the wind fields and the model fields. This is undesirable for the parameter estimations where we expect the error in the observed fields to be related to the parameter perturbations. This is particularly

relevant for sea ice DA where the wind field contribute to most of the uncertainties of the forecast ensemble. Meanwhile, we have also shown that the effect of external uncertainty depends on the model fields. The incorrect cross-correlations is more detrimental when the model fields are directly linked to the source of the uncertainty. For example, when only $\alpha$ is biased, assimilating only SIV shows severely problematic analysis. This may also suggest that coupled DA controlling the uncertainty of external forcing could improve the sea ice parameter estimation.





 ## 7   Conclusions

We investigated the state and parameter estimation in a dynamics-only MEB sea ice model under an idealised setup using the
IEnKF(-N). We mimicked the observation error and its spatial distribution with the forecast uncertainty driven primarily by the
uncertainties in the wind field.

We adopted a fully multivariate approach whereby all model fields are estimated by DA utilising the cross-correlations
between observations and model fields. Our results show that different combinations of the sea ice observation fields can lead
to different effects on the model state and parameter estimates. In general, it is useful to assimilate as many observations as
possible. Potential issues with multivariate state and parameter estimation due to both the limitations of the DA method and
the features of the sea ice model dynamics are highlighted. We also demonstrate that these issues are surmountable and that
successful multi-variate state and parameter estimation using state-of-the-art ensemble DA approaches is possible.

Experiments in the perfect model scenario show that, even if the sea ice model is perfect, the limited ensemble size and
uncertainties in the external forcing, e.g. the atmospheric wind, still limit the capability of DA to improve the sea ice forecast
especially for the unobserved model fields. We show that the forecast of SIV cannot be improved because it is strictly con-
strained by wind field while other model fields with longer timescales show improved forecasts. This suggests that coupled
DA that estimates the external forcing could improve the sea ice forecast of model fields like the SIV. In addition, the linear
Gaussian assumption of DA methods can violate the bounds of various model fields in the sea ice models requiring a post-
processing of these fields. One potential treatment of the bounds problem is anamorphosis which can be applied analytically to
the IEnKF as well as to other EnKF variants (e.g., Bocquet and Sakov, 2013; Simon and Bertino, 2009,  ). We did not explore
this venue here, it may constitute an interesting follow-up work; note however that the majority of sea ice DA does not use
anamorphosis since it can cause numerical imbalances (Bocquet and Sakov, 2013). Nevertheless, with suitable localisation and
sufficient observations, we show improvements for all model fields, both observed and unobserved fields (e.g., stress, cohesion
and damage).

We choose two global parameters to be estimated, the air drag coefficient, $C_a$, and the damage parameter, $\alpha$. The air drag
coefficient is closely related to the external uncertainties from the wind field while $\alpha$ affects the mechanical and dynamical
regime of the sea ice. In a model with only one biased parameter, the DA can reduce the parameter bias and improve the model
forecast. However, in the case where only SIV is assimilated, our results show that it can lead to model imbalance. Also, as
it is closely related to the external uncertainties, the cross-correlation between SIV and $\alpha$ can be incorrectly specified. This
shows the importance of assimilating multiple sea ice observations and the potential difficulty in sea ice DA when the ensemble
uncertainties are primarily driven by the external uncertainties (wind field).

When both model parameters are biased, an identifiability problem arises. This highlights the caveats that the ensemble
spread can come from different sources of uncertainties rather than purely from uncertainties in the parameters. When one
source of uncertainty dominates the ensemble uncertainty, the estimated model parameters can deviate from the truth even if
better forecast is achieved for those observed fields. We proposed a strategy that can mitigate such an issue in our specific test
case by first controlling the dominating error from the external forcing and estimating the sea ice internal parameters later.





A number of open questions are still at stake. For example, we observed improvements in the fully multivariate update but it

is still unclear whether these improvements can be observed in the full Arctic sea ice predictions compared to state-of-the-art operational setup. Another point that should be addressed in future work is the development of a rigorous approach to handle bounded variables like concentration and damage.

On the side of the ensemble generation, in this study, the wind forcing serves as the sole source of uncertainty. Nevertheless, in operational sea ice models, multiple potential uncertain external forcing sources are present. A study of MEB-like models

sensitivity to different external forcing appears as another venue worth pursuing, along the line of the sensitivity analysis to external wind and cohesion parameter in Cheng et al. (2020).

*Code and data availability.* The code for the data assimilation scheme and experiment setup can be found at https://zenodo.org/record/ 8224997. The dynamics-only sea ice model is available upon request.

## Appendix A: Model parameters

Here we present the model parameters used in the modelled truth in Table A1.

## Appendix B: Wind field

The wind field is prescribed as a series of passing cyclonic storms combined with a constant background wind. To have control over the vorticity and divergence of the wind field we use the Helmholtz decomposition

$$\mathbf{u}_a = \nabla \boldsymbol{\Phi} + \mathbf{k} \times \nabla \boldsymbol{\Psi}, \tag{B1}$$

where $\mathbf{u}_a = (u_a, v_a)$, $\boldsymbol{\Phi}$ is the velocity potential and $\boldsymbol{\Psi}$ is the streamfunction with the solenoidal wind component $\mathbf{u}_a^s = \nabla \boldsymbol{\Phi}$ and the divergent wind component $\mathbf{u}_a^d = \mathbf{k} \times \nabla \boldsymbol{\Psi}$. The equations for the wind field is

$$
\begin{aligned}
u_a^s(\hat{x}, \hat{y}, t) &= U_s \sin^2(2\pi\hat{x}) \sin(4\pi\hat{y}) \cdot \gamma(t) \cdot \mathrm{U_{mask}}, & (\text{B2}) \\
v_a^s(\hat{x}, \hat{y}, t) &= -U_s \sin^2(2\pi\hat{y}) \sin(4\pi\hat{x}) \cdot \gamma(t) \cdot \mathrm{U_{mask}} + \mathrm{U^b}, & (\text{B3}) \\
u_a^d(\hat{x}, \hat{y}, t) &= U_d \sin^2(2\pi\hat{y}) \sin(4\pi\hat{x}) \cdot \gamma(t) \cdot \mathrm{U_{mask}}, & (\text{B4}) \\
v_a^d(\hat{x}, \hat{y}, t) &= U_d \sin^2(2\pi\hat{x}) \sin(4\pi\hat{y}) \cdot \gamma(t) \cdot \mathrm{U_{mask}}, & (\text{B5})
\end{aligned}
$$

with $U_d = 0.1 \ \mathrm{m \cdot s^{-1}}$ as the magnitude of the divergent wind and $U_s = 22 \ \mathrm{m \cdot s^{-1}}$ as the maximum solenoidal wind speed, and $U^b = 2 \ \mathrm{m \cdot s^{-1}}$ as the background wind field. To simulate a sequence of storms generating and dissipating, we introduce the time dependent term, $\gamma(t)$, that controls the strength of the storm

$$\gamma(t) = 1 - \exp\left(-a \left| \sin\left(\pi \frac{t}{T}\right) \right|^b\right), \tag{B6}$$





**Table A1.** The default model parameters used in the modelled truth.

| Parameter | Notation | Value |
|---|---|---|
| Poisson's ratio | $\nu$ | 0.3 |
| Internal friction coefficient | $\mu$ | 0.7 |
| Ice density | $\rho$ | $900\,\mathrm{kg\,m^{-3}}$ |
| Elastic (shear) wave propagation speed | $c$ | $500\,\mathrm{m\,s^{-1}}$ |
| Damage parameter | $\alpha$ | 4.0 |
| Undamaged elastic modulus | $E^0$ | $2c^2(1+\nu)\rho$ |
| Undamaged relaxation time $\left(\frac{\eta^0}{E^0}\right)$ | $\lambda^0$ | $10^7\,\mathrm{s}$ |
| Undamaged apparent viscosity | $\eta^0$ | $\lambda^0 E^0$ |
| Minimum apparent viscosity | $\eta_{min}$ | $10^4\,\mathrm{Pa\,s}$ |
| Minimum cohesion | $C_{min}$ | $5000\,\mathrm{Pa}$ |
| Model time step | $\Delta t$ | $30\,\mathrm{s}$ |
| Mean model resolution | $\Delta x$ | $15\,\mathrm{km}$ |
| Characteristic time for damage | $t_d$ | $\Delta t$ |
| Characteristic time for healing | $t_h$ | $5 \times 10^5\,\mathrm{s}$ |
| Air density | $\rho_a$ | $1.3\,\mathrm{kg \cdot m^{-3}}$ |
| Air drag coefficient | $C_{d_a}$ | $1.5 \times 10^{-3}$ |
| Water density | $\rho_w$ | $1027\,\mathrm{kg \cdot m^{-3}}$ |
| Water drag coefficient | $C_{d_w}$ | $5.5 \times 10^{-3}$ |
| Parameter used in coupling $E$ and $\eta$ to $A$ | $c^*$ | 20 |

where $T$ is the period of the storm, and $a = 10$ and $b = 2$ are parameters that control the shape of the curve and in turn the rate that the storm generates, maintains its maximum strength, and then dissipates. In Eq. B2- B5, the spatial coordinate, $(\hat{x}_t, \hat{y}_t)$, is time-dependent. This simulates the passing of the storm across the domain.

$$\hat{x}_t = \hat{x}_{t-1} + \Delta\hat{x}_t, \tag{B7}$$

$$\hat{y}_t = \hat{y}_{t-1} - v_c\Delta t/L, \tag{B8}$$

where $\Delta t$ is the forcing frequency, $v_c$ is the speed at which the storm passes across the domain (from the bottom to top), $L = L_{max} - L_{min}$ and $\Delta\hat{x}_t$ is a random perturbation that allows us to send the storm on a random walk. The initial position of each storm is given by

$$\hat{x}_0 = \frac{(x - x_c) - L_{min}}{L} - \frac{1}{4}, \tag{B9}$$

$$\hat{y}_0 = \frac{(y - y_c) - L_{min}}{L} - \frac{1}{4}, \tag{B10}$$

where $(x_c, y_c)$ is the initial position of the storm centre. The choice of $T$, $v_c$, and $x_c, y_c$ is given in Table 2, and the $\Delta\hat{x}_t$ related to random walk is specified in Eq 4.



With these setup, each storm is approximately half the width of the domain. The sine and cosine equations can generate 4 storms on the domain, and $U_{mask}$ is used to ensures only one storm is in the domain at any time:

$$U_{mask} = (1 - \lfloor x \rfloor - 2 \left\lfloor \frac{\lfloor x \rfloor}{2} \right\rfloor) H(\sin(2\pi x) H(\sin(2\pi y), \qquad \text{(B11)}$$

where $[]$ is the integral part function and $H(x)$ is a Heaviside step function.

*Author contributions.* YC conducted the experiments, performed the data analysis, and wrote the paper. PS developed the wind field perturbation. AC, IP, LB, VD contribute to conceptualization of the work, to the discussion and to the paper writing. MB contributed with the DA codes and the writing. This paper builds on VD's MEB model. TF and YC modified IO routines and boundary conditions for the purposes of simulations. All authors have contributed to the experiment's design, interpreting the results, the discussion, and writing the paper.

*Competing interests.* The authors declare that they have no conflict of interest.

*Acknowledgements.* The authors acknowledge the support of the Scale-Aware Sea Ice Project (SASIP) funded by Schmidt Futures – a philanthropic initiative that seeks to improve societal outcomes through the development of emerging science and technologies. YC has also been funded by the UK Natural Environment Research Council LTS-S award NE/R016518/1. CEREA is a member of Institut Pierre-Simon Laplace (IPSL).





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
