# Peer review of "Multivariate state and parameter estimation with data assimilation on sea-ice models using a Maxwell-Elasto-Brittle rheology"

_EGUsphere, 2023_

## Author Comment (AC1)

**Reviewer 1**

The authors estimate two parameters in a sea-ice model by means of state-space augmentation with an EnKF. The application is clear and useful. However, I have several concerns about the experiment setup. I recommend publishing the manuscript, after the authors have considered my comments below.

**General comments:**

I am concerned about how the authors chose to use their computational resources. I am not asking that the authors execute all the simulations I suggest below, but perhaps they can explain their choices a bit better.

1. Have the authors done a sensitivity ensemble free run in which each member has different parameter values drawn from the assumed distribution, but share the same initial and boundary conditions? From this, one can determine for which values of the parameter the model variables are most sensitive, as well as establish which obs are important for the estimation of the respective parameters, thereby reducing the number of experiments with all the different combinations of obs. See also specific comment 4.

   **Answer:** Thanks for raising the point which is important and that we might not have clarified enough. The choice of the parameters to be estimated is based on previous studies. Massonnet et al. (2014) showed that a coupled ocean-sea ice model is sensitive to the drag coefficient, that is estimated using the ensemble Kalman filter. Using neXtSIM, an MEB rheology-based model such as the one used in our current study, Rabatel et al. (2018) showed the importance of the wind stress, and thus the air-drag coefficient, in determining the spread of the sea-ice trajectories; Cheng et al. (2020) furthermore demonstrated that the shape and orientation of the area covered by the sea-ice trajectories depended on the drag coefficient. Besides the drag coefficient, the damage parameter is the only other minimal parameters in a solid-like rheology-based model, and they are the sole two tunable parameters present in the MEB model used in this study.

   In the new version of the manuscript we have further clarified the rationale for choosing those parameters, but, prompted by the Reviewer's remark, have also added a numerical sensitivity analysis which shows that the observations are sensitive to both parameters. Nevertheless, perturbing $\alpha$ leads to limited change in the ensemble spread in the free run that, in line with our interpretation of our DA results, might be the result of large uncertainty from the wind that dominates the ensemble spread.

   In Sect. 3.1 of the revised manuscript, we added the following text:

   *The choice of the parameters to be estimated is largely based on previous studies. Massonnet et al. (2014) showed that a coupled ocean-sea ice model is sensitive to the drag coefficient, that is estimated with the ensemble Kalman filter. Using neXtSIM, a rheology-based model similar to the MEB model used here, Rabatel et al. (2018) showed the importance of the wind stress, and thus the air-drag coefficient, in determining the spread of the sea-ice trajectories. Furthermore, Cheng et al. (2020) demonstrated that the shape and orientation of the area covered by the sea-ice trajectories depend on the sea-ice air drag coefficient. Besides the drag coefficient, the damage parameter is the only other minimal parameters in a solid-like MEB rheology-based model. Drag coefficient and damage parameters are the sole two tunable parameters present in the MEB model used in this study. This is because, in a dynamics-only MEB model, the air drag coefficient and the damage parameter are the only two parameters that do not affect the maximum speed of the fastest propagating elastic waves which influence the model stability with given time steps.*

   *We have performed a numerical sensitivity analysis which shows that the observations are sensitive to both parameters. Results are shown in Fig. 1 where the same initial and boundary conditions are used while each ensemble member has different $C_a$ or $\alpha$. The parameter values are sampled from $\mathcal{N}(2.5 \times 10^{-3}, 5 \times 10^{-4})$, and $\mathcal{N}(6, 1.5)$ for $C_a$ and $\alpha$ respectively. The model parameters are time-independent, and the ensemble means are $2.5 \times 10^{-3}$ and $6.5$ respectively. Figure 1 shows that both $C_a$ and $\alpha$ have a strong impact on SIC and SIT while a lesser effect is observed on the SIV.*

2. The authors aim to improve long term forecasts by estimating the model parameters. Yet, no long term forecast was computed. On which timescales does the benefit of the parameter estimation persist? And how is the spread/skill ratio for those forecasts?

   **Answer:** Thank you for raising this point. Following the Reviewer's remark, we added the short Sect. 5.5 with the following content:

   *It is worth studying the impact of the parameter estimation on the long-term performance of the model. To this end, we performed $90-days$ long free run whereby the model parameters are "frozen" to the values obtained at the end of the DA period. The experiment is carried out for the case when both $C_a$ and $\alpha$ are estimated during the DA period using the configuration "seq" of Sect. 5.4. Results are displayed in Fig. 2. Figure 2 shows that the RMSE of all observed model variables is smaller using the corrected model parameters compared to the free run that use the initial guess. The improvement is persistent (in time) in SIV, while it increases for SIC and SIT due to the long time scale of these model variables. These results demonstrate the importance of correctly specified model parameters in long-term sea ice forecast.*

3. How does the parameter estimation do compared to perfect model experiments? Because of the different techniques for inflation, one cannot compare. It therefore feels like two separate papers: one for state estimation only, and one for parameter estimation, without a clear connection. Have the authors considered running a perfect model experiment with the same inflation technique as for the parameter estimation experiments? I also wonder about the effect the different inflation techniques have.

[Figure]

Figure 1: Sensitivity of SIV, SIC and SIT to perturbed $C_a$ or $\alpha$ parameter. The shaded area is the standard deviation, a representation of uncertainty of the ensemble, an indication of the strength of the sensitivity.

**Answer:** We thank the Reviewer for their remark. In the updated manuscript, we rerun the experiments using the same adaptive inflation technique across all of the experiments, those with state estimation only (perfect model scenario) and those with simultaneous state and parameter estimation. The new results are qualitatively the same as before albeit minor differences caused by the inflation techniques exist. Our previous overall interpretation of the parameter estimation is thus confirmed by the new results. We have modified the description of the experiments and of the results based on the new inflation strategy. In particular the adaptive inflation appears to bring improved model state analysis in the case of joint parameter estimation. The adaptive inflation also leads to a better uncertainty estimation of the model parameters as we no longer need to overly inflate the ensemble spread of parameters as needed in the previous experiments.

4. I fail to understand the added value of repeating all experiments with all the different combinations of obs for all four cases. For example, it is clear a priori that SIV is important for $C_a$. Why run 3 experiments without SIV? To confirm the importance of SIV one should be enough (or none even). I think the computational resources could be better spent on verifying the significance of the results, since the results look somewhat noisy. For example, how does the parameter estimation do with a different realisation of the truth? Ideally one would estimate as best as possible a distribution for the parameter. In this case a Gaussian distribution $\mathcal{N}(\mu_c, 5 \times 10^{-4})$ for $C_a$ and $\mathcal{N}(\mu_\alpha{}^\prime, 0.5)$ for $\alpha$. Then draw randomly from that distribution to get the true parameter and draw from the same distribution to get the initial ensemble. Repeat this many times for statistical significance. I understand that this may be too expensive, so, as an alternative, one could pick at least a few values for the truth. For example, $+\sigma$, $+2\sigma$, $-\sigma$ and $-2\sigma$. Or other values based on sensitivity studies (see the first general comment). Basically, how wrong does the model have to be to get improvement from the parameter estimation.

**Answer:** Thank you for suggestions. There are two major reasons for comparing different type of observations: 1) due to the limitation and accessibility of sea ice datasets, many operational centres only use SIC, although they are actively pursuing the assimilation of SIT in their operational systems. For example, Zuo et al. (2019) only uses SIC observations in the ECMWF operational systems; Similarly, it is only recent that the UK Met Office experiments with the assimilation of SIT data (Fiedler et al., 2022; Mignac et al., 2022). The SIV is comparatively unexploited. See Table 6.2 in Alvarez Fanjul et al. (2022) for a recent overview. 2) different observations are also a way to show the sensitivity of parameters to observation errors. Tests with different observations also expose potential problems in parameter estimations. For example, in the case of estimating $\alpha$ when only SIV is observed, our results demonstrate the impact of errors from wind forcing interfering with the parameter estimation. Remarkably, our experiments for inferring $C_a$, indicate that even without observing SIV we can achieve good estimations.

Thank you for the suggestions on using different realisations of truth and we fully agree that the manuscript would gain statistical significance. To achieve this goal, extensive investigations with different model state and forcing as well as parameter values should be carried out. Nevertheless, even though the model is a simplification of a full pan-Arctic sea ice model, we lack the computational resources for a satisfactory statistically sound analysis.

With the aforementioned computational constraints in mind, we have however followed the Reviewer's suggestion and have performed a few additional experiments using different truths.

[Figure]

Figure 2: RMSE and ensemble spread of the free run using prior parameters and parameters corrected by the DA when both $C_a$ and $\alpha$ are erroneous.

We added the following paragraphs in Sect. 5.4 of the manuscript: *To further consolidate our findings on the performance of the IEnKF on the simultaneous parameter estimations of $C_a$ and $\alpha$, extensive, and computationally demanding, investigations with different model states, forcing and parameters would be ideal. In fact, even though the model is a simplification of a full pan-Arctic sea ice model, the computational power quickly scales up to beyond our resources.*

*With the aforementioned computational constraints in mind, we performed three additional experiments using different truths. In addition to the existing experiment ($EXP\alpha^L C_a^L$) where we assimilate all observations to estimate $\alpha$ and $C_a$, we implemented three additional experiments where the truth uses 1) $\alpha = 4$ and $C_a = 3.5 \times 10^{-3}$ ($EXP\alpha^L C_a^H$); 2) $\alpha = 7$ and $C_a = 1.5 \times 10^{-3}$ ($EXP\alpha^H C_a^L$); 3) $\alpha = 7$ and $C_a = 3.5 \times 10^{-3}$ ($EXP\alpha^H C_a^H$). Here, the superscript L and H denotes that the truth is lower and higher than the initial guess respectively. We investigate relevant scenarios (e.g. under or overestimate of the real values) in which the model's qualitative behaviour is of the same sort. The latter concerns specifically to the model structural stability, that is to say to the fact that model is not subject to bifurcation of its general behaviour.*

*When the truth of model parameters is higher than the initial guess, the air drag coefficient is $2\sigma$ above the initial guess while $\alpha$ is one $\sigma$ above the initial guess. Keeping the value up to one $\sigma$ higher than the initial guess of $\alpha$ is due to the changes in the dynamical regime of the model that occur beyond one $\sigma$. As discussed in Dansereau (2016), when $\alpha$ increases, the sea ice loses memory of the previous damage events leading to increasing elasto-plastic behaviour. This implies that a different set of initial guess might be needed for these dynamical regimes as we do not expect IEnKF can provide reliable estimation when a change of dynamical regime occurs. Meanwhile, based on physical arguments, such high $\alpha$ parameter is less likely to occur in reality where the true value is likely to be between 4 and 7. As shown in Fig. 3, $EXP\alpha^H C_a^L$ gives improvements in the air drag coefficient, and, though overestimated, the $\alpha$ parameter is increased as prescribed by the truth. In $EXP\alpha^L C_a^H$, improved $C_a$ estimation is obtain at the start of the estimation but the estimation deteriorates after 20 days. In $EXP\alpha^H C_a^H$, though the $C_a$ parameter gets improved, the estimation of $\alpha$ is approximately 17 which is one order of magnitude larger than the truth. The deteriorated results occur when the true $C_a$ is higher than the initial guess which corresponds to a strong wind forcing in the truth. One possible explanation is that the correlations between $\alpha$ and the observed fields are not truthfully reflected due to the strong wind forcing. Nevertheless, if we first estimate the air drag coefficient, followed by a free forecast phase and estimate $\alpha$ afterwards, the parameter estimation is improved. These show that even if the same prior distribution is used, different dynamical regime of the modelled truth can lead to different results in the experiments.*

In response to the concern about the noisy time series of estimated parameters, even though only four experiments were conducted, to obtain a general view of the performance of IEnKF on the parameter estimation in the MEB sea ice model, we present their mean RMSE and standard deviation as shown in Fig. 4. In general, it shows that IEnKF can improve the $C_a$ estimation but lead to deteriorated $\alpha$ estimation. The "sequential" approach marginally improves the $\alpha$ estimation but reduces the benefits of IEnKF when estimating $C_a$. Due to the limited sample size, we do not present these results in the manuscript.

We added above discussion at the end of Sect. 5.4. In the conclusion, we also added the following sentence: *In this study, we choose one specific setup of the model parameters as the truth based on experiments in Dansereau (2016). With our chosen model setup, ....* As part of the future work, we also added: *For example, the MEB model contains multiple dynamical regimes as the*

[Figure]

Figure 3: Simultaneous parameter estimation of $C_a$ and $\alpha$ with different set of truth of these parameters. The horizontal red lines represent the truth when they are either lower or higher than the initial guess. The inset in a) is the estimated $\alpha$ in the experiment $EXP\alpha^H C_a^H$.

[Figure]

Figure 4: RMSE and standard deviation/uncertainty of the RMSE (shaded) over the four experiments for the simultaneous parameter estimations of $C_a$ and $\alpha$.

[Figure]

Figure 5: Variable localisation is used for multivariate state estimation.

*sea ice can be viewed as different materials based on the state of sea ice. It is of theoretical interest to investigate the parameter estimations under different dynamical regimes of the MEB model.*

5. Based on the inset of Figure 3a, the discussion on violation of SIC bounds when SIT is observed, and the paragraph starting on page 17, line 391: Instead of running the SIC+SIT30+SIV experiments, have the authors considered to not allow SIT measurements to affect SIC (and maybe damage)? Basically setting the covariances between SIT and SIC (and damage) to zero only for SIT measurements.

**Answer:** Thanks for the suggestion. The new results are presented in Fig. 5. We added the experiment in the manuscript and added the following discussion:

*Another approach to avoiding the negative impact of SIT consists in artificially setting to zero the covariance matrix entries between SIT and SIC/damage. This is achieved in practice by assimilating only SIC and SIV for the SIT and damage variables while all observations are assimilated for the rest of the state vector. The experiment "var loc" shows improved results in Fig. 5.*

**Specific comments:**

1. page 4, line 99: Why the additional constraint?

   **Answer:** We add the following explanation in the revised manuscript:

   *The latter additional constraint has been included on the basis of theoretical arguments and numerical experiments. It avoids overfitting at a single analysis step for parameters without time-dependency. Without the additional stopping criterion, the parameter estimation leads to excessive corrections that do not appear in the case of state estimation only. The use of early stopping criteria in the context of ensemble-variational methods with domain localisation was also originally suggested by Bocquet (2016) to deal with the potential convergence problems.*

2. page 7, line 187: What does DA-only mean in this context?

   **Answer:** We apologies for the typo. The sentence now reads:

   *The operational sea ice DA corrects only the model state without dealing with model errors explicitly.*

3. Section 5.1: I suggest moving this section to section 4.3.2, as it is meant as tuning for the experimental setup. Naturally, it is up to the authors.

   **Answer:** Thank you for the suggestion. We now moved the original Section 5.1 as part of Section 4.3.2.

4. Table 5. How much does the mean change? Based on the low increase in spread for $\alpha$, I am surprised the estimation works so well. I am guessing that the mean of certain variables changes significantly when changing the parameters? See also general comment 1.

   **Answer:** Thanks for raising the question. A sensitivity study was carried out in response to the first major comment. As suggested by Fig. 1, the observations are sensitive to $\alpha$ but not as much as they are to the air drag coefficients. This is the reason for the failure of assimilation of SIT when estimating $\alpha$ in Sect. 5.3 where the error from the external wind forcing dominates and the DA algorithm is not able to identify and correct the source of errors.

5. For Figures 4, 7, 12 and 14, I suggest using a colour bar that goes as far in red as blue. I understand that the red would be barely visible, but I think that is a good thing, because it would reflect that the deterioration of the fields is significantly less than the improvements.

   **Answer:** Thank you for raising this point. As requested, we adjust the colour bar of the figures.

[Figure]

Figure 6: Distributions of the correlations between different pairs of model variables when all observations are assimilated. The values are taken from all spatial points and analyses steps.The legend indicate the standard deviation of the distributions. Larger spread/standard deviation of the histogram data represents an overall higher correlations between variables.

6. page 17, line 387: I think the statement needs to be reversed: improvement in SIC and SIT (SIT) when SIV (SIC) is assimilated. Also, I am confused about the statement about the comparison between Figure 3 and 4. The two figures seem to be consistent for the chosen localisation radius except when assimilating SIT only. In figure 4 it seems to have a neutral effect on SIC, in contrast to the inset of Figure 3a.

    **Answer:** Thanks for raising this point. We think the confusion is due to lack of clarity in the sentences. We rephrase the sentence as the following:

    *Figure 5 also shows almost no changes/slight improvements in SIV analysis when SIC alone is assimilated and conversely for the SIC analysis when SIT is assimilated. This appears in contrast to Fig. 4, which indicate that SIT observations have a negative impact on the SIC analysis, and that SIC observations can deteriorate SIV analysis at certain localisation radii (cf. Fig. 4a and 4b insets)*

7. Table 7: I personally lack intuition about what the actual values mean in Table 7. Perhaps small histograms of correlation values or small scatter plots would be useful here.

    **Answer:** We understand the Reviewer's concern and have substituted the original Table 7 with a figure that we hope will better serve our purposes. Our motivation is to show, and to a certain degree quantify, why the DA is in general (i.e. average over the full spatial domain and across all analyses) having across-variables effects. To this end we display the distributions of the cross correlations estimated by the IEnKF, those that are then used to do the analysis update. We have modified the text in the manuscript to accommodate and describe the new figure as per the following:

    *We show the strength of the cross-correlations between different model variables in Fig. 6. The values are taken from the experiment where all observations are assimilated, from all spatial points and all analyses. As expected, the distributions are all peaked around zero: this is because beyond a certain distance, the correlations are all very small (a fact that is at the basis of the use of localisation), with the larger values concentrated in the proximity of the analysis point and populating mainly the tales of the distributions in Fig. 6. The width of the distributions indicates that in many instances the correlations are (in absolute value) as high as 0.5. To provide a quantitative comparison among the distributions' width, we also shows the standard deviations.*

8. Figure 8: The colours of the free run and the forecast are too similar.

    **Answer:** Thank you for noting this. The colour of the forecast is changed to black and we hope it is visually better.

9. page 27, line 574: Isn't the reason the estimation works well when alpha is estimated only after 10 days, simply because $C_a$ is already significantly more accurate and therefore the parameter estimation problem converges to another (better) local minimum? In other words, the problem resembles the case 3 scenario.

    **Answer:** Yes. This is exactly the case. The treatment is partly motivated by the fact that we know the dominant uncertainty in the observed model variables comes from wind forcing. When this uncertainty is controlled, IEnKF can correctly control the uncertainty in $\alpha$.

**References**

Alvarez Fanjul, E., Ciliberti, S. A., and Bahurel, P.: Implementing Operational Ocean Monitoring and Forecasting Systems., 2022.

Bocquet, M.: Localization and the iterative ensemble Kalman smoother, Q. J. R. Meteorol. Soc., 142, 1075–1089, https://doi.org/10.1002/qj.2711, 2016.

Cheng, S., Aydoğdu, A., Rampal, P., Carrassi, A., and Bertino, L.: Probabilistic forecasts of sea ice trajectories in the Arctic: impact of uncertainties in surface wind and ice cohesion, Oceans, 1, 326–342, https://doi.org/10.3390/oceans1040022, 2020.

Dansereau, V.: A Maxwell-Elasto-Brittle model for the drift and deformation of sea ice, Theses, Université Grenoble Alpes, URL `https://theses.hal.science/tel-01316987`, 2016.

Fiedler, E. K., Martin, M. J., Blockley, E., Mignac, D., Fournier, N., Ridout, A., Shepherd, A., and Tilling, R.: Assimilation of sea ice thickness derived from CryoSat-2 along-track freeboard measurements into the Met Office's Forecast Ocean Assimilation Model (FOAM), The Cryosphere, 16, 61–85, https://doi.org/10.5194/tc-16-61-2022, 2022.

Massonnet, F., Goosse, H., Fichefet, T., and Counillon, F.: Calibration of sea ice dynamic parameters in an ocean-sea ice model using an ensemble Kalman filter, Journal of Geophysical Research: Oceans, 119, 4168–4184, https://doi.org/https://doi.org/10.1002/2013JC009705, 2014.

Mignac, D., Martin, M., Fiedler, E., Blockley, E., and Fournier, N.: Improving the Met Office's Forecast Ocean Assimilation Model (FOAM) with the assimilation of satellite-derived sea-ice thickness data from CryoSat-2 and SMOS in the Arctic, Quarterly Journal of the Royal Meteorological Society, 148, 1144–1167, https://doi.org/https://doi.org/10.1002/qj.4252, 2022.

Rabatel, M., Rampal, P., Carrassi, A., Bertino, L., and Jones, C. K. R. T.: Impact of rheology on probabilistic forecasts of sea ice trajectories: application for search and rescue operations in the Arctic, The Cryosphere, 12, 935–953, https://doi.org/10.5194/tc-12-935-2018, 2018.

Zuo, H., Balmaseda, M. A., Tietsche, S., Mogensen, K., and Mayer, M.: The ECMWF operational ensemble reanalysis–analysis system for ocean and sea ice: a description of the system and assessment, Ocean Science, 15, 779–808, https://doi.org/10.5194/os-15-779-2019, 2019.

---

## Author Comment (AC2)

**Reviewer 2:**

**Summary**

The authors investigate the fully multivariate state and parameter estimation though idealized simulations of a dynamics-only model using the MEB sea ice rheology. They employ an iterative ensemble Kalman Filter (iEnKF) DA approach with a stopping criteria set to 40 iterations. The model runs are performed with a spatial resolution of 15 km and a 30 sec timestep to ensure numerical stability while resolving propagation of damage.

Four scenarios are evaluated inferring the model physical variables 1) under a perfect model setup (truth), 2) and the drag coefficient $C_a$, 3) and its erroneous damage parameters $\alpha$, and 4) and its erroneous $C_a$ and $\alpha$. Different inflation strategies are used for all 4 scenarios. In scenario 1, a 42-day run free ensemble without DA, is followed by a series of 30-day assimilation experiments in which 3 of the 9 fields are bounded quantities (SIC, SIT, level of damage). When only 1 field is assimilated, that field gets most of the improvement. The cross-correlation between differing quantities is examined. In scenario 2, only one observation type (SIC or SIT) is assimilated, the analysis underestimates $C_a$ at the end of the analysis time. However, assimilating SIC was closer to truth. They found the best skill in estimating $C_a$ when assimilating SIV alone due to its close relationship with wind forcing and $C_a$. In scenario 3, the assimilation of SIC or SIT alone led to and under- and over-estimation of $\alpha$ after 30 days. The simultaneous assimilation of SIC and SIT led to almost a full recovery of the true $\alpha$ value of 4. Results showed that observations of SIV can not be used to retrieve $\alpha$ effectively. When all types of observations were assimilated in Scenario 4, the estimate of $C_a$ was furthest from the truth. They found that the forecast of SIV can not be improved because it is strictly constrained by the wind field while other model fields with longer timescales showed improved forecasts. They suggest that coupled DA that estimates external forcing could improve SIV.

This a well written and well referenced paper with clear thought put into designing and executing the experiments. The tables and figures are concise and easy for the reader to understand. I recommend publication with minor edits as outlined below.

**General Comments:**

1. Line 395-396: Please comment on the statement "when SIT is assimilated with SIC, the adverse effect is subdued". This is partially true for $D < 0$, but for $d > 1$, damage is at 20.93%. Please explain.

   **Answer:** Thanks for the comment. This is a valid point. We add the following discussion on this in line 396 (revised manuscript):
   *Interestingly, however, when SIT is assimilated together with SIC, the boundedness of the level of damage is improved for undamaged sea ice ($d < 0$ for analysis before post-processing) but is not so for completely damaged sea ice ($d > 1$ for analysis before post-processing). Yet, it is sufficient to improve the overall RMSE of the level of damage (see Fig. 5). One possible reason is that, without the thermodynamics, the forecast error mainly comes from the damaged sea ice and the overestimation of undamaged sea ice has little contribution to the RMSE after the post-processing .*

**Specific Comments:**

1. Line 45: Add the following references: (Xie et al., 2018, Blockley and Peterson (2018), Fiedler et al., 2022)

   Blockley, E. W. and Peterson, K. A.: Improving Met Office seasonal predictions of Arctic sea ice using assimilation of CryoSat-2 thickness, The Cryosphere, 12, 3419–3438, https://doi.org/10.5194/tc-12-3419-2018, 2018.

   Fiedler, E. K., Martin, M. J., Blockley, E., Mignac, D., Fournier, N., Ridout, A., Shepherd, A., and Tilling, R.: Assimilation of sea ice thickness derived from CryoSat-2 along-track freeboard measurements into the Met Office's Forecast Ocean Assimilation Model (FOAM), The Cryosphere, 16, 61–85, https://doi.org/10.5194/tc-16-61-2022, 2022.

   **Answer:** They are added, thanks for the suggestions.

2. Line 54: with "the" changing number…

   **Answer:** Added, thanks.

3. Line 64: such "a" model

   **Answer:** Added, thanks.

4. Line 70: …has not yet "been" studied extensively…

   **Answer:** Added, thanks.

5. Line 87: …DA system "consisting" of

   **Answer:** Changed, thanks.

6. Line 96: rephrase to "methods can be found in chapter 7"

   **Answer:** Changed, thanks.

7. Line 105: remove "we"

   **Answer:** Removed, thanks.

8. Line 153: ..."the" DA's ability...

   **Answer:** Added, thanks.

9. Line 164: ...with "a" uniform...

   **Answer:** Added, thanks.

10. Line 396: effect is subdued check...

    **Answer:** This sentence is removed due to the major comment.

11. Figure 8: "true" and "forecast" blue lines are a bit difficult to differentiate. Can one of the colors be changed?

    **Answer:** Thanks for raising the point. We changed the colour of the forecast line from light blue to black, the truth uses a red line and the freerun still uses a blue line. We hope it reads better visually.

12. Line 593: use of "the" ensemble OR use of "an" ensemble

    **Answer:** Added, thanks.

13. Line 617: ...shed "light" (not plural)

    **Answer:** Changed, thanks.

14. Line 642: Is there another reference to add after "Bertino, 2009"? If not, remove the "," before the ")"

    **Answer:** Removed, thanks.

**References**

---

## Author Response (AR2)

**Technical correction**

Please ensure that the colour schemes used in your maps and charts allow readers with colour vision deficiencies to correctly interpret your findings. Please check your figures using the Coblis – Color Blindness Simulator (https://www.color-blindness.com/coblis-color-blindness-simulator/) and revise the colour schemes accordingly. = Figs. 5, 9, 13

    **Answer:** Thank you for the suggestion. We assume that the suggested changes of Figs. 5, 9, 13 were based on the numbering in the original manuscript (the line plots) instead of the revised manuscript (red-blue heatmap). As one of the co-authors is a typical colour blind, he confirmed that the line plot was not sufficiently clear and we revised those figures. He also confirmed that the red-blue heatmap is clear for him.